# Sialic acids in pancreatic cancer cells drive tumour-associated macrophage differentiation via the Siglec receptors Siglec-7 and Siglec-9

Ernesto Rodriguez [1], Kelly Boelaars[1], Kari Brown[1], R. J. Eveline Li[1], Laura Kruijssen[1], Sven C. M. Bruijns[1], Thomas van Ee[1], Sjoerd T. T. Schetters [1], Matheus H. W. Crommentuijn[1], Joost C. van der Horst[1], Nicole C. T. van Grieken[2], Sandra J. van Vliet [1], Geert Kazemier[3], Elisa Giovannetti [4,5], Juan J. Garcia-Vallejo[1] & Yvette van Kooyk [1✉]

Changes in glycosylation during tumour progression are a key hallmark of cancer. One of the glycan moieties generally overexpressed in cancer are sialic acids, which can induce immunomodulatory properties via binding to Siglec receptors. We here show that Pancreatic Ductal Adenocarcinoma (PDAC) tumour cells present an increased sialylation that can be recognized by Siglec-7 and Siglec-9 on myeloid cells. We identified the expression of the α2,3 sialyltransferases ST3GAL1 and ST3GAL4 as main contributor to the synthesis of ligands for Siglec-7 and Siglec-9 in tumour cells. Analysing the myeloid composition in PDAC, using single cell and bulk transcriptomics data, we identified monocyte-derived macrophages as contributors to the poor clinical outcome. Tumour-derived sialic acids dictate monocyte to macrophage differentiation via signalling through Siglec-7 and Siglec-9. Moreover, triggering of Siglec-9 in macrophages reduce inflammatory programmes, while increasing PD-L1 and IL-10 expression, illustrating that sialic acids modulate different myeloid cells. This work highlights a critical role for sialylated glycans in controlling immune suppression and provides new potential targets for cancer immunotherapy in PDAC.

[1] Amsterdam UMC, Vrije Universiteit Amsterdam, Department of Molecular Cell Biology and Immunology, Cancer Center Amsterdam, Amsterdam Infection and Immunity Institute, Amsterdam, Netherlands. [2] Amsterdam UMC, Vrije Universiteit Amsterdam, Department of Pathology, Amsterdam, Netherlands. [3] Amsterdam UMC, Vrije Universiteit Amsterdam, Department of Surgery, Cancer Center Amsterdam, Amsterdam, Netherlands. [4] Amsterdam UMC, Vrije Universiteit Amsterdam, Department of Medical Oncology, Cancer Center Amsterdam, Amsterdam, Netherlands. [5] Cancer Pharmacology Lab, AIRC Start-Up Unit, Fondazione Pisana per la Scienza, Pisa, Italy. ✉email: y.vankooyk@amsterdamumc.nl

Pancreatic ductal adenocarcinoma (PDAC) remains one of the most severe cancer types with a 5-year survival of 9%[1]. PDAC is characterised by early invasion to adjacent tissue and metastasis that hampers the possibility for surgical resection of the tumour, which is the only potentially curative treatment for PDAC patients[2]. Other therapies such as chemotherapy, targeted therapy and immunotherapy fail to improve survival of patients, which is mainly attributed to the aggressive and complex tumour microenvironment (TME)[3]. In order to find new treatment targets for PDAC, identification of novel immune suppressive networks in the TME is urgently needed.

The PDAC TME is characterised by dense fibrotic stroma and suppressive immune cells, mostly of the myeloid lineage with only few CD8[+] T cells[3]. The myeloid compartment is composed of a heterogeneous and multifaceted group of cells that can have different impact on tumour progression. For example, dendritic cells (DCs) can activate anti-tumour cytotoxic T cells to restrain tumour growth, while some types of macrophages and immature myeloid cells can promote tumour progression and immune evasion[4]. Macrophages are generally chronically polarised in the TME to have tumour-promoting properties and are referred to as tumour-associated macrophages (TAMs). Polarisation of TAMs goes beyond the M1/M2 paradigm as they express markers of both inflammatory M1 macrophages (HLA-DR) and tissue repair M2 macrophages (CD206, CD163) and are also functionally involved in both inflammation and tissue remodelling[5,6]. Both TAMs and DCs in the TME can originate from infiltrating monocytes[7,8]. Yet, little is known on the factors that determine the local differentiation of infiltrating monocytes to TAMs or DCs.

It is well established that tumours present an altered glycosylation machinery that facilitates tumour progression[9]. Glycosylation is a posttranslational process that occurs in the endoplasmic reticulum (ER) and Golgi apparatus during the biosynthesis and transportation of proteins and lipids to the cell membrane[10]. In this process, monosaccharides are added in a stepwise manner to the carrier on proteins or lipids, which is regulated by the local presence or absence of glycosyltransferases, glycosidases and sugar donors[10]. These glycans can be sensed by immune cells via glycan binding receptors called lectins. Increasing evidence illustrates that tumour cells alter their glycosylation patterns and overexpress 'self-associated patterns' that trigger immune-inhibiting lectin receptors[9].

The changes in glycosylation during the development of PDAC, which we define as glyco-code, are not fully defined. PDAC patients have elevated levels of the serum marker CA19-9, corresponding to Sialyl Lewis[A], which is currently used as disease biomarker in diagnostics[11]. Other described glycosylation-related changes in PDAC are upregulation of sialyl Lewis[X], truncated O-glycans, galectin-1 and galectin-3, specific proteoglycans and increased branched and fucosylated N-glycans, which are all associated with distinct aspects of tumour progression such as tumour cell proliferation, invasion, metastasis, inflammation and angiogenesis[12]. However, their link to immune cell function in PDAC is unknown.

Enhanced sialylation (hypersialylation) is frequently observed in solid tumours[13]. Sialic acids can be recognised by a family of receptors called Siglecs (Sialic acid-binding immunoglobulin-type lectins), most of which possess an *immunoreceptor tyrosine-based inhibitory motif* (ITIM)[14]. Upon engagement with Siglec receptors, sialylated glycans can trigger tolerogenic programs in different immune cell types, such as T cells, NK cells and monocytes[14]. In mouse models, the presence of sialic acids on tumour cells has been associated with the induction of regulatory T cells (Tregs) and a reduction in effector T cells, and increased tumour growth[15]. By binding DCs, sialylated antigens have also been shown to induce a regulatory phenotype by promoting IL-10 secretion and Treg induction, illustrating that tolerizing pathways are induced upon binding of sialic acids[16]. However, still little is known on how local sialic acid expression connects to Siglec expression and the induction of tolerogenic programs on immune cells in the PDAC TME.

In this paper, we show that PDAC tumour cells present increased sialylation that is sensed by the myeloid receptors Siglec-7 and Siglec-9, contributing to the differentiation of monocytes into macrophages with an immune-suppressive phenotype. In conclusion, we find a link between the presence of sialic acids in the TME that modulate monocyte and macrophage behaviour, associated with worse clinical outcomes.

## Results

**PDAC tumour cells show enhanced expression of α2,3 linked sialic acids.** Sialic acid metabolism involves a series of enzymes responsible for the synthesis of CMP-sialic acid, which is the donor later used by different sialyltransferases to add sialic acids to an extending glycan structure (Fig. 1A). These glycoconjugates can present sialic acids in different linkages with respect to the underlying glycan (namely α2,3, α2,6 and α2,8), each of which are catalysed by specific enzymes (Fig. 1A)[14].

To characterise changes that occur in the sialylation machinery of PDAC, we analysed publicly available transcriptomic data that contained samples of both PDAC and normal or adjacent normal tissue (Supplementary Table 1)[17–20]. To investigate which specific sialylation pathways are differentially expressed between tumour and normal tissue, we used gene set enrichment analysis and differential gene expression (Fig. 1B, Supplementary Fig. 1A). We found that the sialic acid-donor synthesis pathway is particularly upregulated in PDAC, observing an increased expression of *GNE*, *NANP* and *SLC35A1*, suggesting a general increase of sialylation in the tumour (Fig. 1B, Supplementary Fig. 1A). Moreover, pathways associated with the synthesis of α2,3 and α2,6 sialylated glycans are increased (Fig. 1B). The genes that were mostly upregulated are *ST3GAL1* and *ST6GALNAC1*, which encode the enzymes responsible for the synthesis of sialyl-T and sialyl Tn antigen, respectively, known to be upregulated in pancreatic cancer (Supplementary Fig. 1A)[21].

To validate the increased expression of sialic acids in PDAC observed in the transcriptomic analysis, we performed immuno-histochemistry on biopsies from PDAC patients using the biotinylated plant lectins derived from *Maackia amurensis* (MALII) and *Sambucus nigra* (SNA). These lectins can be used as probes to identify sialylated structures with the α2,3 or α2,6 configuration, respectively (Fig. 1C). Quantification of the signal intensity in the ductal cell, revealed that an increased sialylation in cancer cells respected to normal ducts (Fig. 1C, Supplementary Fig. 1B). Sialic acid-containing glycans can be found also in the stroma of PDAC, which may be secreted by tumour cells or derived by other cells of the tumour microenvironment (Supplementary Fig. 1C).

We further analysed the expression of α2,3 and α2,6 linked sialic acids on different PDAC cell lines. In an ELISA setting, we found that lysates from primary and commercial PDAC cell lines contain α2,3 and α2,6 sialic acids, while a normal-like pancreatic ductal cell line (HPNE) only presented α2,6 sialylated structures (Fig. 1D). However, when assessing the surface expression of sialic acids using flow cytometry, we found that all PDAC cell lines showed a high expression of α2,3 sialic acids, while only a few cell lines expressed α2,6 sialic acids (Fig. 1E). Since α2,3 sialic acids are consistently expressed on the surface of PDAC cells, and are differential expressed between the normal duct and PDAC

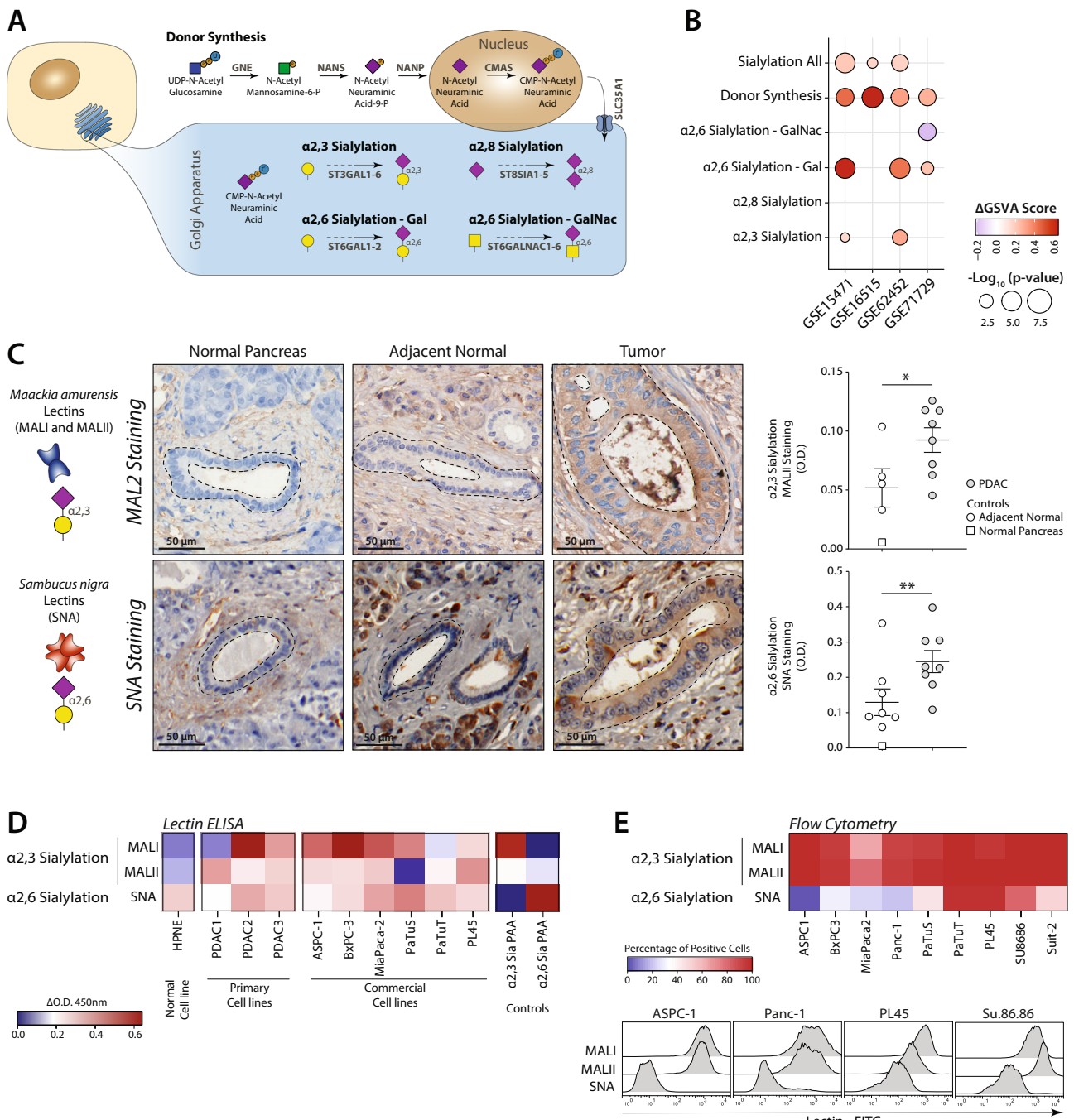

**Fig. 1 Sialylation is increased in pancreatic ductal adenocarcinoma (PDAC). A** Representation of the different pathways that contribute to sialylation of glycans. **B** Gene set enrichment analysis of the pathways described in (**A**) in normal and tumour tissue. ΔGSVA score was calculated as the difference between the GSVA score in tumour and in normal tissue. **C** Immunohistochemistry analysis of the expression of sialylated glycans in normal and tumour tissue, using plant lectins specific for α2,3 (MAL I and MAL II) and α2,6 sialic acid (SNA). Data presented as mean values ± SEM. **D–E** Evaluation of sialic acid expression in PDAC cell lines by (**D**) ELISA and (**E**) flow cytometry. ΔD.O. at 450 nm was calculated as the difference of the O.D at 450 nm of the sample and the one of the uncoated control.

ductal cells, which is not the case for α2,6 sialic acids, we focussed on regulation and recognition of α2,3 sialylated structures.

**Sialic acids present on PDAC cells are recognised by Siglec-7 and Siglec-9 present on myeloid cells.** Given that sialylated structures can serve as ligands for Siglec receptors, we next evaluated whether PDAC cell lines could be recognised by a panel of different Siglec-hFc chimeras. Interestingly, we could only detect ligands for Siglec-7 and Siglec-9, which were able to

interact with tumour cells in a sialic acid-dependent manner, as its removal by treating the cells with neuraminidase, a glycosidase specific for sialic acid, reduced their binding (Figs. 2A, S2A). Also, in lysates from primary and commercial PDAC cell lines, Siglec-7 and 9 ligands were detected in ELISA setting (Fig. 2B). Siglec ligands are consistently expressed on the surface of PDAC cells in a similar pattern of α2,3 sialic acids, which is not the case for α2,6 sialic acids (Figs. 1E, 2A). These data suggest that α2,3, rather than α2,6, play a key role in sialic acid–Siglec interactions.

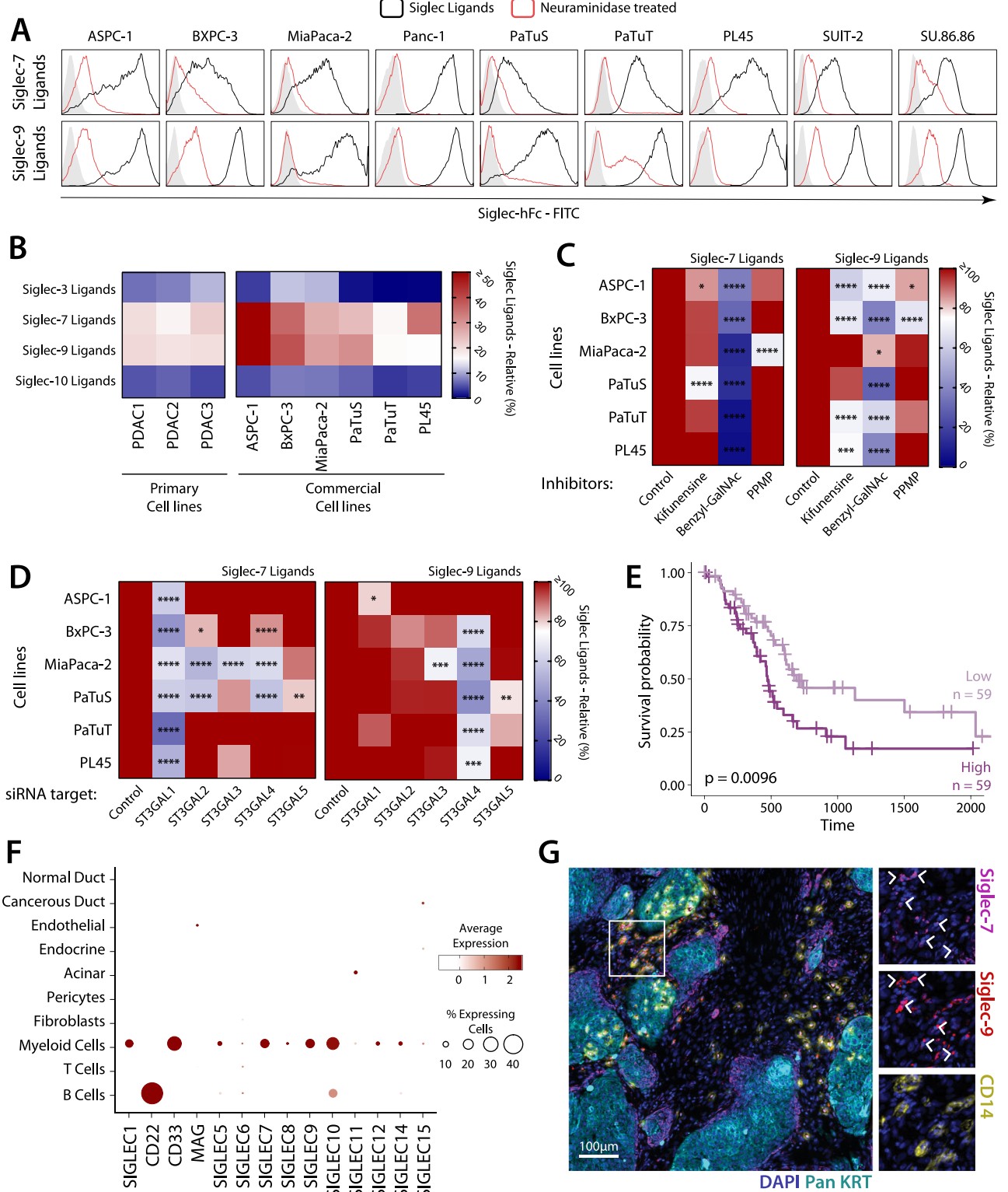

**Fig. 2 PDAC cancer cells express ligands for the myeloid receptors Siglec-7 and Siglec-9. A–B** Expression of Siglec-7 and Siglec-9 ligands in PDAC cancer cell lines evaluated by flow cytometry (**A**) and ELISA (**B**). In ELISA, the positive control for each Siglec-hFc was set as 100%. Representative results of three independent experiments. **C**, **D** Characterisation of Siglec ligands by using (**C**) glycosylation inhibitors and (**D**) knock down of different a2,3 sialyltransferases. Mean of control siRNA was set as 100%. Statistics: Pairwise comparisons of each condition against the respective control using Two-way ANOVA with Dunnett's multiple comparisons test (*$p \leq 0.05$; **$p \leq 0.01$; ***$p \leq 0.001$; ****$p \leq 0.0001$). **E** Survival analysis of PDAC patients based on the mean expression of *ST3GAL1* and *ST3GAL4* using the Log-rank test in the TCGA-PAAD data set. Top and bottom thirds of the mean expression were used to define high and low expression, respectively. **F** Expression of Siglec receptors in different cell populations present in PDAC tumours in scRNA-Seq from Peng et al. **G** Immunofluorescent staining of Siglec-7 and Siglec-9 receptors in PDAC. Representative staining of $n = 6$ PDAC patients.

To characterise the ligands for Siglec-7 and Siglec-9 in the PDAC cell lines, we next analysed whether they were presented in N- or O-linked glycosylated proteins or were presented on glycolipids. Accordingly, we treated the cell lines with different glycosylation inhibitors that include Kifunensine, which inhibits mannosinase I enzyme thereby blocking complex N-glycan synthesis; Benzyl-GalNAc, which inhibits the extension of O-glycans and PPMP, an inhibitor of glycolipid synthesis (Supplementary Fig. 2B). The sialic acid-containing glycans for Siglec-9 were found to be N- and O-linked, whereas sialic acid ligands for Siglec-7 were mainly restricted to O-glycans (Fig. 2C).

Subsequently, we analysed which enzymes contribute to the synthesis of Siglec-7 and Siglec-9 ligands in pancreatic cancer cell lines, by using siRNA to knock down the expression of several sialyltransferases, focussing on those that catalyse the synthesis of α2,3 sialylated structures. We identified ST3GAL1 as a main contributor to the synthesis of Siglec-7 ligands and ST3GAL4 participating in the synthesis of Siglec-9 ligands (Fig. 2D). ST3GAL1 has been reported to mainly catalyse the sialylation of the T-antigen, a short O-glycan product often increased in tumour, in agreement with our results with glycosylation inhibitors. ST3GAL4 is described to catalyse multiple glycosidic processes, which may explain the presence of Siglec-9 ligands in both O- and N-glycans. The combined expression of ST3GAL1 and ST3GAL4 was associated with significantly shorter survival of PDAC patients (Fig. 2E). Interestingly, we did not observe differences in survival with the genes individually, probably mutual compensation between the ligands of Siglec-7 and Siglec-9. Overall, these results suggest that Siglec-7/9 ligands may be increased in PDAC biopsies and may promote tumour progression.

Expression of Siglec-7/9 receptors has been reported on myeloid cells, NK cells and some subsets of T cells[14,22,23]. To investigate which immune cell populations in PDAC can sense the cancer-derived sialic acids via Siglec-7/9, we analysed single-cell RNA sequencing (scRNA-seq) data of human PDAC biopsies and control pancreas, previously published by Peng et al.[24]. Our results confirm the presence of the ten different populations defined by Peng et al. and showed an increase of myeloid cells and T lymphocytes in PDAC (Supplementary Figs. 3A, B). Interestingly, Siglec expression is mainly restricted to myeloid cells, with the exception of the B-cell, associated Siglec-2 (Fig. 2F). However, Siglecs were not detected in other cell populations, including T cells (Fig. 2F). It must be noted that in scRNA-Seq not all the transcripts are detected in all the cells, therefore the percentages depicted only serve as a semi-quantitative measure. Accordingly, when analysing transcriptomic from bulk tissue, the expression of both Siglec-7 and Siglec-9 strongly correlated with markers found in the myeloid population (TYROBP, FCER1G, C1QR and CD14) and were co-expressed on CD14+ myeloid cells in PDAC biopsies (Figs. 2G, S3C, S3D). These results show that tumour sialylation can be sensed by Siglec-7 and Siglec-9 expressing myeloid cells in the PDAC microenvironment.

**Monocyte-derived macrophages are main contributors to poor survival probability in PDAC.** Since Siglec-7/9 are expressed on myeloid cells, we next performed a deep characterisation of the myeloid cell composition in PDAC analysing the scRNA-Seq data, published by Peng et al.[24]. For this, cells within the myeloid cluster were selected, re-normalised and re-clustered. Five clusters were identified that corresponded to *monocyte-derived macrophages* (moMac), *monocyte-derived dendritic cells* (moDC), *tissue-resident macrophages* (trMac)*, monocytes and proliferating macrophages* (Fig. 3A). While most myeloid cells in normal pancreatic tissue are trMac (median of 93,4%), PDAC is largely infiltrated with moMac (36.6%), moDC (27.2%) and monocytes

(9.2%) (Fig. 3A). Less than 5% of myeloid cells were identified as proliferating macrophages which were also enriched in PDAC.

Each cluster identified was assigned to their respective population based on the expression of well established myeloid markers as obtained in the differential gene expression analysis (Fig. 3B, Supplementary Data 1). For instance, moMac was identified by the expression MARCO, CD68 and CCL2 in that specific cluster, which are commonly use macrophage markers. Similarly, the cluster of moDCs showed higher expression of the DCs-specific genes HLA genes, CD74, CD1c and CLEC10A; monocytes were identified by the genes VCAN, S1008 and S100A9 and proliferating macrophages had high expression of the proliferation marker MKI67 and other cell cycle-associated genes (Fig. 3B). In the single-cell RNA sequencing data, both trMac and moMac showed expression of several macrophage markers, as APOE, CD163 and CD206. trMac was identified as tissue-resident by their presence in normal tissue. We also observed the expression of organ specific gene signatures in this cluster, such as insulin and amylase (Fig. 3B). To confirm the proposed classification, we performed enrichment analysis of gene sets derived from previously published transcriptomic data of in vitro generated moDCs and moMac and study their correspondence with the different myeloid populations identified in PDAC (Supplementary Fig. 4A, Supplementary Data 2)[25]. This analysis showed a clear association of the clusters identified Monocytes, moDCs and moMac with gene signatures of their in vitro counterparts (Supplementary Fig. 4A).

All myeloid populations, including monocytes, showed expression of both Siglec-7 and Siglec-9 (Fig. 3C). We therefore continued analysing the differences between patients in their composition of myeloid cells and its association to survival. We identified four different subgroups of PDAC patients based on the composition of their myeloid compartment, characterised by either the abundance of trMac, MoDC, MoMac or a mixture of these cells (Figs. 3D, S4B). As expected, the percentage of moDCs negatively correlated with percentage of moMac, suggesting that the tumour microenvironment may direct infiltrating monocytes to differentiate into either moDC or moMac (Fig. 3E). To study if these differentiation pathways can be found in patients, we used diffusion maps, a dimensional reduction tool that allowed us to analyse trajectories during cell differentiation using single-cell data. Indeed, diffusion maps of moMac and moDC showed clear differentiation pathways from monocytes to either moMac or moDC in PDAC tumours (Fig. 3F).

We next analysed how each myeloid population affected clinical outcome, by defining specific gene sets for each of them and using them to perform gene set enrichment analysis and survival analysis in large bulk transcriptomic data sets (Supplementary Fig. 4C, D, Supplementary Table 1)[26–28]. Interestingly, the presence of moDCs generally correlated with a longer survival, while moMac enrichment defined a more aggressive phenotype (Supplementary Fig. 4D). To evaluate how the relative presence of macrophages over DCs affects the clinical outcome, we defined a moDC/moMac ratio as the difference between the moMac and moDC scores. The stratification of patients based on this score showed that patients with a higher presence of macrophages over DCs had shorter survival (Fig. 3G, Supplementary Fig. 4D). Other myeloid populations did not correlate with survival (Supplementary Fig. 4D). Taken together, these data show that PDAC is characterised by enhanced myeloid cell infiltration and that differentiation of infiltrating monocytes to moMac is associated with shorter survival.

**Tumour cells drive the differentiation of monocytes to moMac.** We next aimed to investigate what drives the differentiation of monocytes to moMac in PDAC. We found that tumours in which

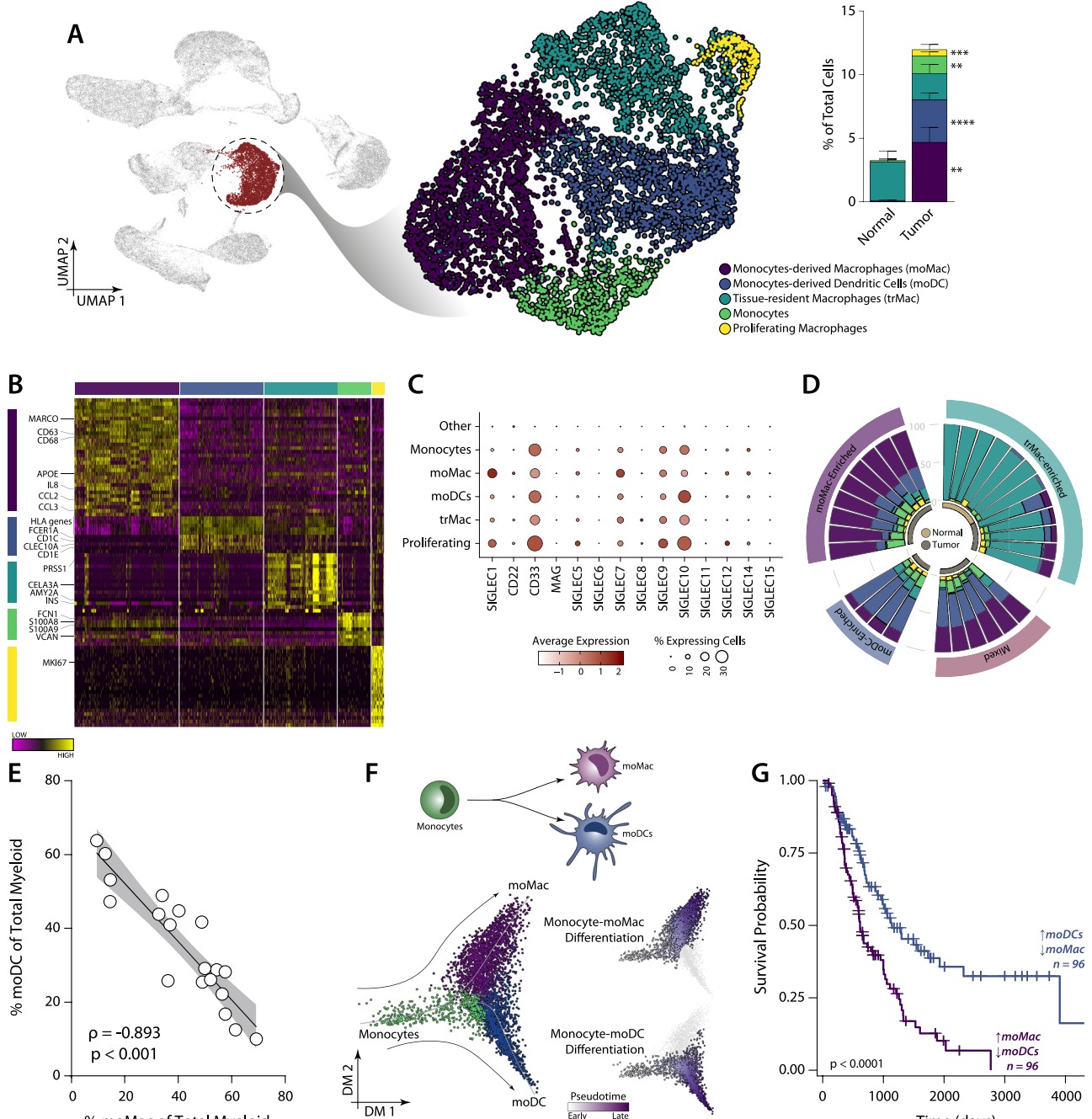

**Fig. 3 Characterisation of the myeloid cells in PDAC. A** Characterisation and quantification of the myeloid cell population found in the scRNA-Seq analysis of PDAC tumours from Peng et al. Data presented as mean values ± SEM. **B** Heatmap highlighting the markers characterising each myeloid population. **C** Expression of Siglec receptors in the different myeloid populations. **D** Clustering of patients using the presence of myeloid populations distinguishes patient groups. **E** Correlation between the moDC and moMac as a % of total myeloid cells. Spearman correlation coefficient, $p$ value and 95% confidence interval are depicted in the figure. **F** Analysis of the differentiation of monocytes towards dendritic cells or macrophages using diffusion map (using the R package *destiny*) and pseudotime (*slingshot*). **G** Survival analysis based on a moMac/moDCs ratio Score in bulk transcriptomic data using the Log-rank test. moMac/moDC ratio Score was defined as the difference between a moMac and a moDCs Score ($\text{Score}_{\text{moMac}} - \text{Score}_{\text{moDCs}}$). Top and bottom thirds were used to define samples enriched in moMac or moDCs, respectively. Data set shown: E-MTAB-6138.

the myeloid compartment is enriched in moMac, had a higher percentage of cancer cells compared to tumours that present a higher amount of moDCs (Fig. 4A). Indeed, we found a positive correlation between the fraction of moMacs in the myeloid compartment and the number of cancer cells (Supplementary Fig. 4E). In line with this result, TAMs have been associated with tumour progression and proliferation in PDAC[8,29]. On the other hand, DCs have a stronger potential to induce anti-tumour

responses, which could explain the lower amount of tumour cells in the moDC-enriched patients. However, this correlation is lost when we consider the amount of moMac in the total sample (Supplementary Fig. 4F). This suggests that the presence of cancer cells is associated with an enhanced differentiation of monocytes towards macrophages and not to the absolute amount of moMac. In this context, we hypothesise that cancer cells are the main drivers of the differentiation of moMacs in PDAC (Fig. 4B). To

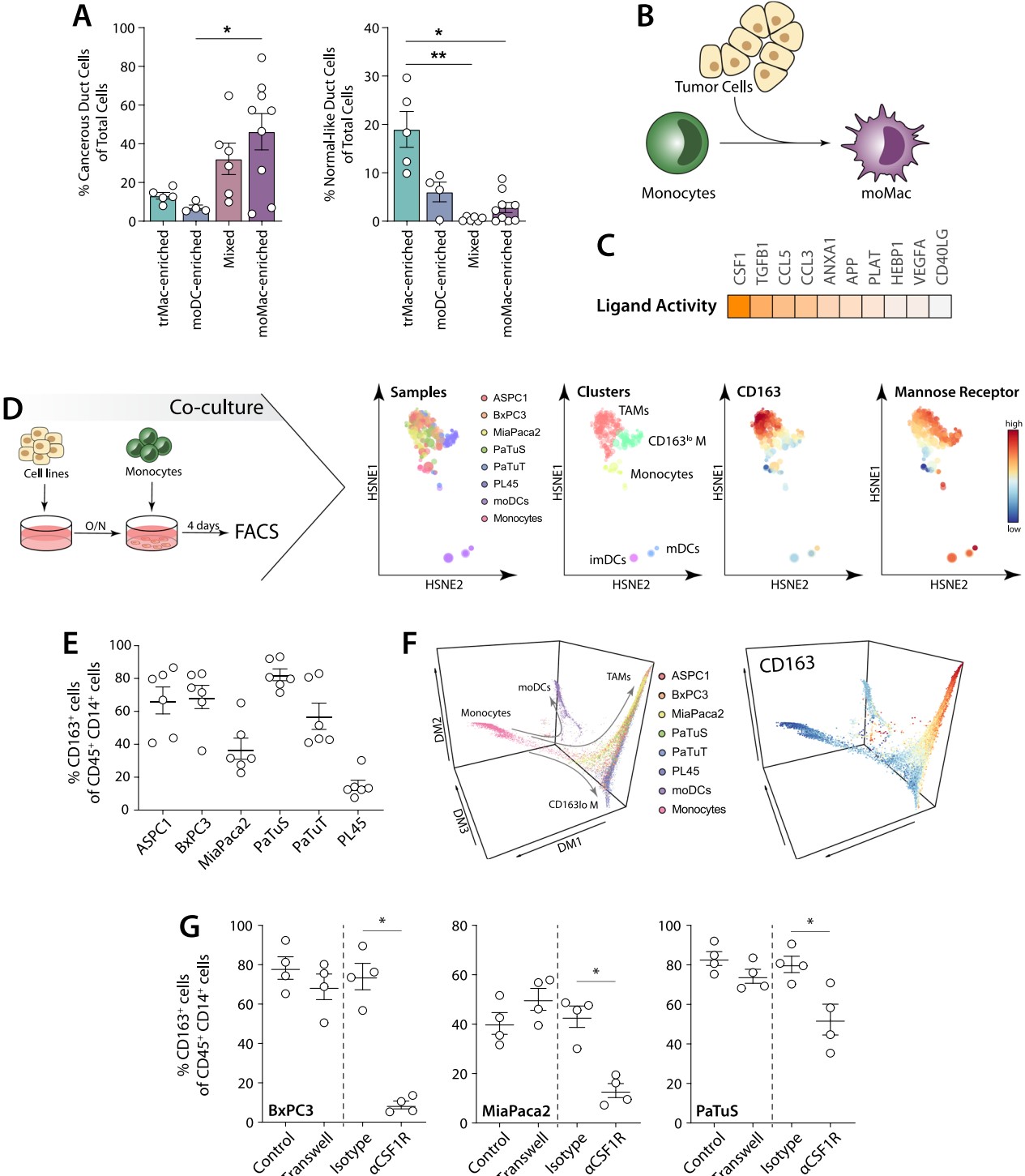

**Fig. 4 Characterisation of the myeloid cells in PDAC. A** moMac-enriched patients contain a higher content of cancer cells in scRNA-seq data from Peng et al. Patients numbers: trMac-enriched ($n = 5$), moDCs-enriched ($n = 4$), mixed ($n = 6$), moMac-enriched ($n = 9$). Data presented as mean values ± SEM. Statistics: Kruskal-Wallis test with Dunn's multiple comparisons test. **B** Proposed model for tumour cell-mediated monocyte to macrophage differentiation in PDAC. **C** NicheNet algorithm predicts M-CSF (CSF1) as tumour-derived factor that influences moMac differentiation. **D** Co-culture of PDAC cell lines with monocytes induces their differentiation to macrophages. **E** Quantification of CD163 positive cells after co-culture indicated in (**D**). Data presented as mean values ± SEM. **F** Analysis of monocyte differentiation using diffusion maps (using the R package *destiny*). Three-dimensional graph showing the inferred differentiation pathways and the expression of CD163. **G** Differentiation of monocytes to CD163[+] macrophages in co-cultures of monocytes and cancer cell lines in a transwell model and in the presence of a blocking antibody to CSF1R. Data presented as mean values ± SEM. Statistics: Friedman test (*$p < 0.05$).

examine which tumour-derived factors are able to drive the monocyte to moMac differentiation, we used the NicheNet algorithm to infer which ligand-receptor pairs are involved (Figs. 4C, S5A)[30]. This analysis showed that tumour-derived M-CSF (gene name *CSF1*) is the main contributor to moMac differentiation (Figs. 4C, S5A).

To further investigate how PDAC cells regulate monocyte differentiation we set up a co-culture system in which we cultured six different pancreatic tumour cell lines with human blood-derived monocytes for four days, and subsequently analysed for multiple macrophage and DC markers using flow cytometry (Fig. 4D). HSNE analysis of the CD45+ immune cells allowed us to identify two distinct populations induced by the PDAC cell lines: a CD163+ CD206+ CD14+ cell population (phenotype generally associated to TAMs) and a population of CD163lo CD206+ CD14+ cells (which we called CD163lo myeloid cells) (Fig. 4D). The PL45 PDAC cell line induced CD163lo myeloid cells, while all other tested cell lines induced TAMs (Fig. 4D-E). To study whether the CD163lo myeloid cells and TAMs represent different stages of differentiation, we analysed the flow cytometry data using diffusion map, which suggested that these cells represent two independent cell types (Fig. 4F). A deeper phenotyping also showed the expression of myeloid markers in TAMs, including HLA-DR, CD68 and CD304 and the lectin receptors CD206 (also called Mannose receptor), DC-SIGN, MGL and CD169 (Supplementary Fig. 5B, C). When compared to in vitro differentiated DCs and macrophages, PDAC-induced TAMs behaved in a similar way to cells generally associated with a M2 phenotype (M-CSF macrophages stimulated with IL-4 and IL-6) regarding their immune-phenotype and response to LPS stimulation (Supplementary Fig. 5C, D). Differentiation to TAMs was induced by secreted factors of the PDAC cell lines, as a transwell system induced similar TAM differentiation as the co-culture (Fig. 4G). Additionally, blocking of CSF1R reduced the differentiation to TAMs, indicating that in vitro the PDAC-induced differentiation to TAMs is mediated by M-CSF, as predicted in silico (Fig. 4G). However, blockade of CSF1R receptor did not fully block the differentiation to TAMs, indicating that other secreted factors also contribute to monocyte-to-TAM differentiation.

**Tumour cell derived sialic acids contribute to TAM differentiation**. We next evaluated whether secreted sialylated glycans contribute to the monocyte-to-TAM differentiation. We found that Siglec-7 and Siglec-9 ligands were also present in the supernatant of tumour cell lines, as assessed by ELISA (Fig. 5A). After validating that blood-derived monocytes express Siglec-7/9 (Fig. 5B), we performed the co-culture as described before in the presence or absence of Siglec-7/9 blocking antibodies (Fig. 5C). Interestingly, blockade of Siglec-7 and Siglec-9 reduced the differentiation to CD163+ macrophages (Fig. 5C). To further evaluate the contribution of sialic acids to TAM differentiation, we created a knockout (KO) PDAC cell line for the Sia-activating enzyme CMP-sialic acid synthase (CMAS), the enzyme that generates the sialic acid donor used by sialyltransferases to generate sialylated structures (Fig. 5D). KO of CMAS in the PDAC cell line BxPC3 led to complete loss of sialic acids, while mock-transfected cells still expressed sialylated molecules on the cell surface (Fig. 5E). CMAS KO PDAC cell lines were able to differentiate monocytes to TAMs that express CD163; however, these TAMs had reduced expression of CD206 (Fig. 5F). Importantly, the co-culture with the CMAS KO tumour cells showed reduced IL-10 production (Fig. 5G). These results show that tumour cell expressed sialic acids can attribute to TAM differentiation via Siglec-7 and Siglec-9, by inducing immune

suppressive properties such as CD206 expression and IL-10 production.

**Activation of Siglec-9 induces a tolerogenic programme multiple myeloid cells**. To evaluate whether sialic acids alone could also induce differentiation to TAMs, we generated dendrimers containing terminal α2,3 sialic acids. When monocytes were incubated with the sialic acid dendrimers, they significantly upregulated PD-L1, IL-10 and IL-6 expression, and showed increased expression of CD163 and CD206 (Fig. 6A-C). No expression of IL-1β, IL12p40 nor TNFα was increased with respect to the medium control. This suggests that α2,3 sialic acids induce a TAM phenotype in the monocytes. However, there is a clear contribution of M-CSF in increasing TAM differentiation, while α2,3 sialic acids alone increased IL-10, IL-6 and PD-L1 expression. When monocytes were differentiated with M-CSF in the presence of α2,3 sialic acid dendrimers, the differentiated macrophages showed increased CD163 and CD206 expression, as well as increased IL-6 production, indicating that α2,3 sialic acids contribute to the differentiation of monocytes to macrophages with immune suppressive properties similar to tumour-associated macrophage (Fig. 6A-C). Surprisingly, when analysing tyrosine phosphorylation of an array of immune receptors after stimulation of monocytes with α2,3 sialic acid dendrimers, we observed that only Siglec-9 showed increased phosphorylation, illustrating that Siglec-9 activation is contributing to the sialic acid induced immune modulation (Fig. 6D-E).

We next investigated whether α2,3 sialylated dendrimers can polarise already differentiated macrophages to TAMs. In vitro generated moMac, moDCs and TAMs showed expression of Siglec-9 and can therefore bind the α2,3 sialic acid dendrimers (Fig. 6F). Similar to the effect on monocytes, dendrimers induced PD-L1 expression and IL-10 expression in M-CSF differentiated macrophages (moMac) (Fig. 6G-I). In addition, α2,3 sialylated dendrimers reduced the expression of HLA-DR and CD86, markers associated with the anti-tumour M1 phenotype of macrophages (Fig. 6H). Moreover, when exposed to the pro-inflammatory stimulus Lipopolysaccharides (LPS), sialic acid-containing dendrimers further reduced the secretion TNFα, illustrating that even under inflammatory conditions the sialic acid/siglec axis is able to modulate immune responses (Fig. 6I). No differences were found in the production of IL-1β and IL12p40. Together, these data show that activation of Siglec-9 by α2,3 sialic acids modulates myeloid cell function in multiple stages of monocyte to TAM differentiation to acquire a immunomodulatory phenotype, by upregulating PD-L1, IL-10, IL-6 and CD206, that may facilitate immune evasion in PDAC (Fig. 6J).

**Discussion**

This study was set up to investigate hypersialylation in PDAC and its role in modulating immune cells in the tumour micro-environment. Here, we reveal that tumours from PDAC patients have increased expression of α2,3 sialic acids that is locally expressed as well as secreted into the TME. There, sialic acids modulate monocytes to produce IL-10 and IL-6 and subsequently drive the polarisation and differentiation of monocytes to immune suppressive TAMs via the activation of the Siglec-9 receptor. This work highlights a mechanism of immune suppression by PDAC cells that could be exploited for cancer immunotherapy.

Glycosylation profiles of cancer cells are strongly associated with tumour progression and immune modulation[9,31]. Here, using transcriptomic data and tissue stainings, we showed that PDAC is characterised by an increased sialylation. Increased

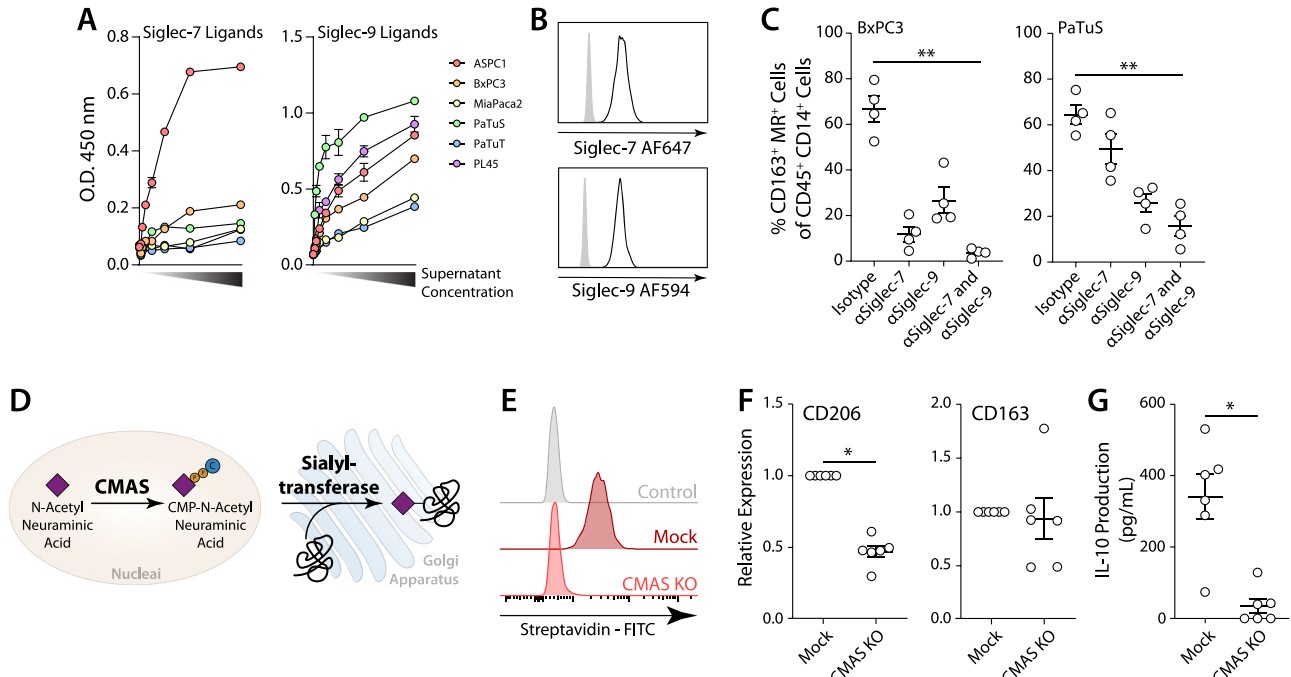

**Fig. 5 Sialic acids contribute to the differentiation of tumour moMac by interacting with Siglec-7 and Siglec-9 in monocytes. A** Presence of Siglec ligands in the supernatants of cell lines analysed by ELISA. **B** Histogram of Siglec-7 and Siglec-9 expression in circulating monocytes as analysed by flow cytometry, non-stained cells in grey and blood-derived monocytes from healthy donors in black line. **C** Analysis of CD163+ MR+ CD45+ CD14+ cells after the co-culture of tumour cell lines with monocytes in the presence of Siglec-7 and Siglec-9 blocking antibodies. Data presented as mean values ± SEM. Statistics: Friedman test (**$p \leq 0.01$). **D** Representation of the role of CMAS in cellular sialylation. **E** Flow cytometric analysis of α2,3 linked sialic acids using MAAII lectin staining in the BxPC3 CMAS knockout and mock cells. **F** Expression of CD206 and CD163 in CD45+ CD14+ cells after co-culture of monocytes with mock and CMAS KO cell lines. Relative expression was calculated with respect to the mock cell lines (set as 1). Data presented as mean values ± SEM. Statistics: Two-tailed Wilcoxon test (*$p < 0.05$). G) IL-10 production after co-culture was analysed by ELISA. Data presented as mean values ± SEM. Statistics: Two-tailed Wilcoxon test (*$p < 0.05$).

sialylation was facilitated by enhanced expression of different enzymes involved in the synthesis of the sugar donor (Sialic acid-CMP), and also some sialyltransferases as *ST3GAL1*, *ST6GAL1*, *ST6GAL2* and *ST6GALNAC1*. Previous studies showed that ST3Gal3 and ST3Gal4 were overexpressed in PDAC which was linked to increased migration metastatic ability of cancer cells[32,33]. ST6Gal1 has also been described to be upregulated in PDAC[34]. Yet, our analysis showed the most pronounced upregulation in the α2,6 sialylation enzyme *ST6GALNAC1*, which is the enzyme that adds a sialic acid to Tn-antigen and creates sialyl-Tn. Tn-antigen is described to be overexpressed in many types of cancer, including PDAC[21]. We also confirmed the increase of sialic acid in tumour cells by immunohistochemistry using the plant lectins SNA and MALII. Interestingly, we also observed the presence of sialic acid-containing glycans in the stroma compartment. These structures may be secreted by tumour cells (as we observed for cell lines) or derived by other stromal cells. In the present paper, we focus on the immunemodulatory properties of tumour-derived sialylated structures, but more research should be performed to determine the particular properties of stroma-derived sialic acid.

Surprisingly, we demonstrate that of all Siglec receptors that can potentially bind sialic acids, only Siglec-7 and Siglec-9 were able to bind sialic acids on the tested cell lines. Studies have shown that recombinant glycosylated MUC-1, in particular MUC-1 with sialylated T-antigen (MUC-ST), can modulate monocyte differentiation via Siglec-9[35]. However, other studies have shown that Siglec-9 does not bind MUC1[36,37]. In addition, Siglec-9 can also bind to sialylated *N*-glycans[38]. Our results also show that Siglec-7 mainly bound sialic acids in O-glycans, and generation of the ligands was mainly controlled by ST3Gal1. The

enzyme ST3Gal1 is responsible for attaching a sialic acid to T-antigen, leading to the generation of sialyl T-antigen[39]. This suggests that Siglec-7 mainly binds to sialyl T-antigen in the context of PDAC, which is in line with previous research showing that MUC1 binds to Siglec-7[37]. On the other hand, our results show that Siglec-9 bound to sialic acids on both O- and N-glycans. The generation of Siglec-9 ligands was regulated by ST3GAL4, which is described to be involved in the generation of sialyl-Lewis[X] antigen, which indeed can be present in both N- and O- glycans[40]. The enzymes involved in the generation of the Siglec-7/9 ligands, ST3Gal1 and ST3Gal4, that are involved in α2,3 sialylation, were both upregulated in PDAC patients, suggesting that upregulation of Siglec7/9 ligands in cell lines is translatable to PDAC patients.

Siglec-7 and Siglec-9 are expressed exclusively in myeloid cells in PDAC. Although the myeloid compartment in PDAC is well characterised in mice models, little is known in human. We here show for the first time the composition of the myeloid compartment in human PDAC, using the public data set published by Peng et al.[24]. Our results show that monocytes that infiltrate the TME in PDAC can differentiate to either DCs or macrophages, in line with our recent report in Melanoma[41]. In mouse PDAC, the pool of TAMs is composed of both monocyte-derived macrophage and tissue-resident macrophages, with the latter being mainly responsible for promoting tumour progression[8]. In contrast, our results reveal that in human PDAC monocyte-derived macrophages, and not tissue-resident macrophages, are the myeloid population associated with worse survival. Our results suggest that monocyte-to-macrophage differentiation is induced mainly by the tumour cells themselves, with M-CSF as a key driver. This was not surprising, as M-CSF is commonly used to

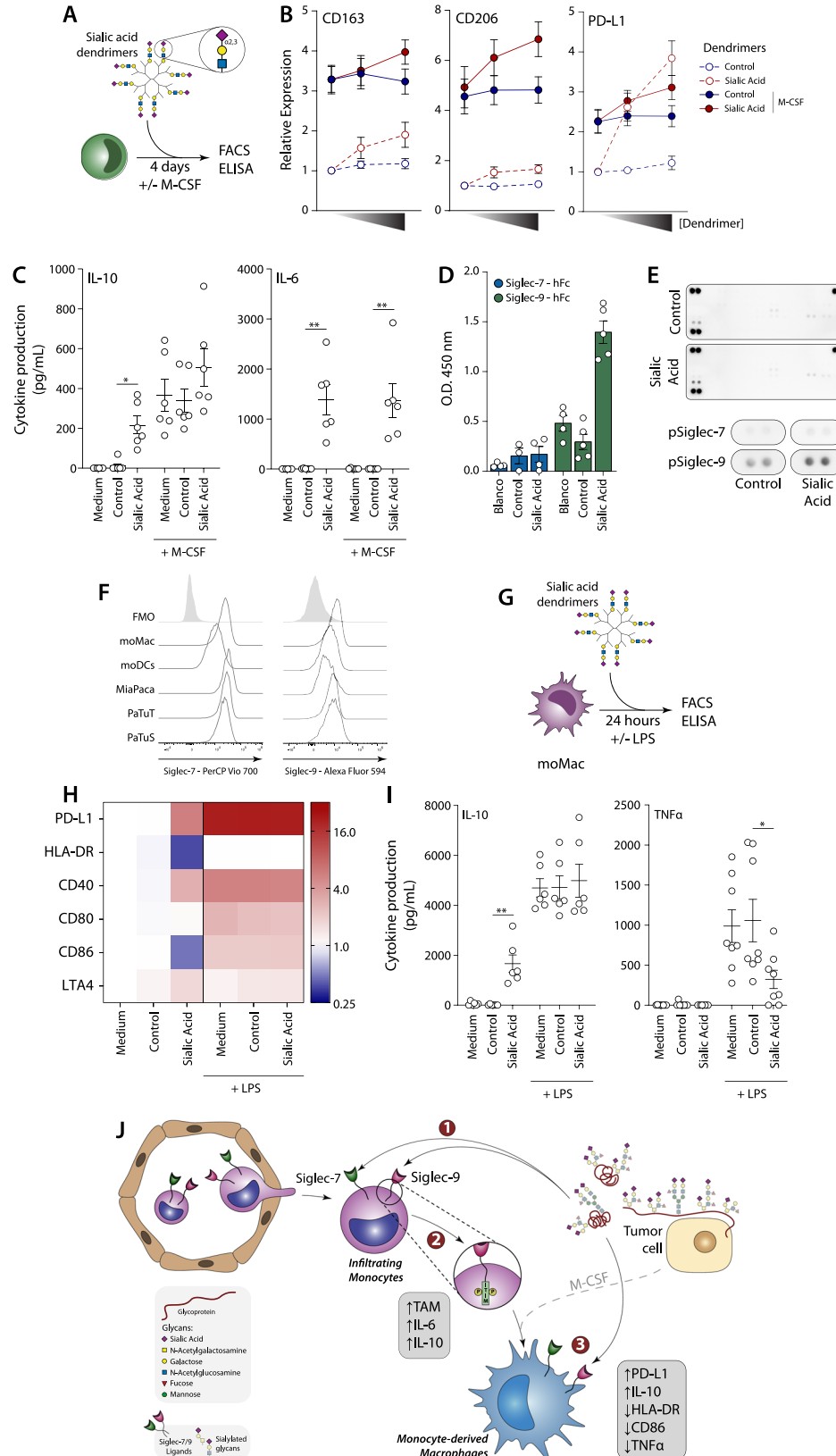

generate macrophages from monocytes in vitro and it has been shown that PDAC cells produce elevated levels of M-CSF compared to normal tissue[42,43]. Targeting the M-CSF/CSF1R axis can reduce tumour progression in multiple murine cancer models including PDAC[29,43–45]. Although several clinical trials have been performed with CSF1R blocking agents, they show disappointing outcomes in solid tumours[46]. On the other side, we recently described that the differentiation of moDCs in the TME favours the efficiency of PD-1 blockade[41]. Given that infiltrating monocytes can contribute to the pool of tumour moMac and moDCs, the blockade of TAM differentiation represents a unique opportunity as complement for immunotherapy. In animal models,

**Fig. 6 Dendrimers carrying a2,3 sialylated structures are able to modulate monocytes and macrophages. A** Monocytes were stimulated with a2,3 sialic acid dendrimers for four days in the presence or absence of M-CSF. **B** Expression of CD163, CD206 and PD-L1 in CD14+ cells after stimulation. Relative expression was calculated with respect to unstimulated monocytes. Data presented as mean values ± SEM. **C** Presence of IL-10 and IL-6 in supernatants was evaluated by ELISA. Data presented as mean values ± SEM. **D** Binding of Siglec-7 and Siglec-9 to sialic acid dendrimers. Data presented as mean values ± SEM. **E** Phospho-Immunoreceptor array of monocytes stimulated with α2,3 sialic acid and control dendrimers. **F** Expression of Siglec receptors in different monocyte-derived macrophages was evaluated by flow cytometry. **G** M-CSF monocyte-derived macrophages were stimulated with α2,3 sialic acid dendrimers in the presence or absence of LPS. **H, I** Expression of co-stimulatory markers (**H**) and cytokine production (**I**) after stimulation. In (**H**), relative expression was calculated with respect to unstimulated macrophages. Statistics: Pairwise comparisons of each condition against the respective control using Two-way ANOVA with Dunnett's multiple comparisons test (*$p \leq 0.05$; **$p \leq 0.01$). **J** Scheme summarising the results of this paper. ① Circulating monocytes that infiltrate the tumour can interact with tumour-derived sialylated structures via the receptors Siglec-7 and Siglec-9. ② Triggering of Siglec receptors synergises with M-CSF to induce the differentiation of moMac and concomitant expression of IL-6 and IL-10. ③ Sialic acid triggering of Siglec-9 in moMac affects their expression of co-stimulatory molecules and cytokine production.

combination of CSF1R blockade and classical immunotherapy agents (as anti-PD1 and anti-CTLA4) has shown a reduced tumour growth compared to single treatment[43]. Here, we clearly observed that the presence of sialic acids during the differentiation of monocytes to TAMs contributes to their anti-inflammatory phenotype and cytokine secretion profile, highlighting its immunomodulatory properties. Thus, removal of sialic acids should be considered when interfering with TAM differentiation in the TME and in immunotherapy strategies.

Sialic acids have been increasingly studied as a target in cancer therapy due to their modulatory properties[9,14]. We have previously described that tumour sialylation is involved in the induction of regulatory T cells and dampens NK activity in animal models[15,16]. Moreover, other researchers showed that local removal of sialic acids in the tumour microenvironment by intratumoural injection of sialic acid mimetics, led to enhanced infiltration of NK and cytotoxic T cells while reducing suppressive myeloid cells[47]. Siglec-9 has also been found on subsets of effector memory T cells within the tumour microenvironment, indicating that removal of sialic acids could also directly influence T cell function[22]. In this paper, we describe that tumour-derived α2,3 sialylated structures drive tumour-promoting monocytes and TAMs via upregulation of CD206, PD-L1 and immunosuppressive cytokines. TAM induction by sialic acid is mediated via the Siglec-7 and Siglec-9 receptor and can be inhibited by blockade of Siglec-7 and Siglec-9. Moreover, the sialic acid was able interfere with the secretion of TNFα after stimulation with the TLR4 ligand LPS, which has been widely used to polarise macrophages towards a pro-inflammatory phenotype (classically known as M1). The re-polarisation of immunomodulatory TAMs in the direction of an M1-like phenotype has been proposed as a therapy for cancer[48,49]. However, the increased sialylation present in the tumour may undermine these efforts if not properly tackled. Given the immune inhibitory motifs of Siglec receptors and their broad distribution on immune cells, the sialic acid-Siglec axis provides a new therapeutic target for immunotherapy.

Targeting the sialic acid-Siglec axis has already shown its promise in mouse models. One strategy uses sialidases targeting HER2 on breast cancer cells, leading to enhanced NK cell mediated tumour killing upon sialic acid removal[50]. Another study described the use of sialic acid mimetics, that incorporate into tumour cells and thereby block attachment of sialic acids to glycans, leading to reduced tumour growth in vivo[47]. Sialic acids on tumour cells can also directly be targeted. Recently, Siglec-7/9 based CAR T cells have been developed that target sialic acid expressing tumour cells, and delay tumour growth in a melanoma model[51]. The present study shows that PDAC cells specifically express Siglec-7 and Siglec-9 ligands, therefore these CAR T cells could potentially be effective in PDAC. In addition, targeting Siglec receptors should be considered. However, murine immune cells express a different array of Siglec receptors than humans,

which hampers pre-clinical evaluation of Siglec blocking therapeutics[9]. Together, we and others illustrate that targeting sialic acids can be a potent immunotherapy. Further research is needed to evaluate its efficacy and safety in humans.

In summary, we identified a mechanism of immune modulation in PDAC where tumour cells induce suppressive myeloid cells via the sialic acid-Siglec axis. The results of this research support the concept that altered tumour glycosylation serves as a novel immune checkpoint[9]. In addition, upregulated Siglec-7 and Siglec-9 ligands could be used as disease biomarkers as we show that these structures are universally upregulated in PDAC. Future research will reveal whether similar mechanisms of immune modulations might occur in other types of cancer.

## Methods

**PDAC patient tissue.** Tumour specimens were obtained from patients undergoing resection at the VU University Medical Center, with the approval from the Local Medical Ethical committee at the VU and the Biobank (#14038). Tumour samples were evaluated for their quality and tumour. Formalin-fixed paraffin embedded (FFPE) tissue was obtained from the Pathology Department of the Amsterdam UMC, location VUMC. Written consent was obtained for all the donors.

**Cell lines.** ASPC1, MiaPaCa-2 and PL45 were acquired from ATCC. BxPC3 is a kind gift from Dr. A. Frampton (Imperial College, London, UK). PaTuS and PaTuT are a kind gift from Dr. I. van Die (Amsterdam UMC, The Netherlands). Cell lines were tested for their authentication by STR-PCR, performed by BaseClear (Leiden, The Netherlands), before the start of the project. Primary cell lines PDAC1-3 were obtained from patients undergoing pancreaticoduodenectomy, as described before[52]. Additionally, all cell lines were routinely tested for Mycoplasma using PCR. All the cell lines were cultured in RPMI 1640 (Gibco) supplemented with 10% Fetal Calf Serum (Biowest), 2 mM L-Glutamine (Gibco) and 1000 U/mL Penicillin-Streptomycin (Gibco).

**Glycodendrimer synthesis.** The generation 2.0 PAMAM dendrimer with a cystamine core (647829, Sigma Aldrich) was conjugated to three different glycans via reductive amination. 20 equivalents of LS-Tetrasaccharide d (LSTd, Elicityl) and D-(+)-Galactose (G0750 Sigma Aldrich) per dendrimer were dissolved in Dimethylsulphoxide (DMSO) and acetic acid (8:2). Per dendrimer, 160 equivalents 2-Methylpyridine borane complex (65421 Sigma Aldrich) was added to a total volume of 200 μL. The reaction was incubated at 65 °C for 2 h with frequent vortexing. The reaction products were purified over disposable PD10 desalting columns (GE17-0851-01 GE Healthcare) in 50 mM Ammonium Formate pH 4.4 ($NH_4HCO_3$).

The presence of the α2-3 sialic acid was validated using an ELISA-type assay using *Maackia Amurensis* Lectin I (MAL-I) (Vector Laboratories, Peterborough, UK). Briefly, NUNC maxisorb plates (RosKilde) were coated overnight at 4 °C with 5 μM of the products. The wells were subsequently blocked for 2 h at room temperature with carbo-free blocking buffer (Vector, SP5040). Incubation with the biotinylated MAL-I and peroxidase-labelled streptavidin (Sigma-Aldrich) allowed spectrophotometric quantification of the binding with 3,3′,5,5′-tetramethylbenzidine (TMB, Sigma-Aldrich) at 450 nm on the iMark™ Microplate Absorbance Reader (Bio-RAD).

**Monocyte and macrophage stimulation.** Buffy coats were obtained from healthy donors (Sanquin, The Netherlands). Peripheral blood mononuclear cells (PBMCs) were isolated by density gradient centrifugation with Ficoll-Paque PLUS (GE Healthcare), to later purify CD14+ monocytes using MACS CD14 MicroBeads (Miltenyi). Monocytes were cultured in RPMI 1640 (Gibco) supplemented with

10% Fetal Calf Serum (Biowest), 2 mM L-Glutamine (Gibco) and 1000 U/mL Penicillin-Streptomycin (Gibco).

To study the capacity of PDAC cell lines to induce the differentiation of monocyte-derived macrophages (moMac), monocytes were co-cultured with cell lines in 24-well plates for 4 days and phenotype was analysed by flow cytometry. *Cytosplore* was used for the analysis of macrophage phenotype by using Hierarchical Stochastic Neighbor Embedding (HSNE)[53]. Differentiation analysis was done using diffusion maps as implemented in the R package *destiny*. For Siglec blocking experiments, monocytes were pre-incubated with neutralising antibodies for Siglec-7 (Biolegend) or Siglec-9 (R&D Systems), with a final concentration of 2.5 μg/mL.

Alternatively, pooled samples from the differentiation induced by different cell lines were sorted in CD45+ CD163+ and CD45+ CD163−/low cells and stimulated with LPS for 24 h. To compare the phenotype with cytokine-induced moMacs, monocytes were stimulated with 50ng/mL M-CSF or 20 ng/mL GM-CSF for 3 days. The polarisation of the M-CSF induced moMac was performed by incubation with 20 ng/mL IL-4, 20 ng/mL IL-6 or 20 ng/mL IFNγ and 10 ng/mL LPS.

To study if sialylated structures can affect macrophage differentiation, monocytes were stimulated for 4 days with 5 μM of sialic acid dendrimers in the presence or absence 20 ng/mL of M-CSF. The phenotype was studied by flow cytometry and supernatants were harvested for the determination of cytokine secretion by ELISA.

For the stimulation of Macrophages with glyco-dendrimers, moMacs were generated by incubation of monocyte with 50ng/mL M-CSF for five days, with a change at day three. 100.000 moMacs were stimulated with 5 μM of sialic acid or control dendrimers in the presence or absence of 10 ng/mL LPS (Sigma-Aldrich). Supernatants were harvested 18 h later for the determination of cytokine secretion by ELISA.

**PDAC tissue staining, imaging and analysis.** Fresh frozen pancreatic tumour sections were cut at 10 μm thickness and following blocking with Carbo-Free Blocking Solution (Vector Labs). Sections were then fixed with a 1% Paraformaldehyde solution and following washing steps, incubated for 16 h at 4 °C with AlexaFluor 594-conjugated anti - Siglec 9 (R&D Systems), AlexaFluor 647-conjugated anti - Siglec 7 (R&D Systems), AlexaFluor 700-conjugated anti - CD14 (Sony) and with AlexaFluor 488-conjugated anti - Pancytokeratin (eBioscience). Tissue sections were counterstained with DAPI (Invitrogen) and mounted in FluoromountG mounting medium (ThermoFisher Scientific). Slides were scanned using the Vectra Polaris (PerkinElmer). Multispectral images (MSIs) were acquired and unmixed using spectral libraries built from single-stained beads (UltraComp eBeads, Invitrogen) or tissue sections for each fluorescent dye. A list of the antibodies used in this paper can be found in Supplementary Table 2. The inForm Advanced Image Analysis Software Version 2.4.2 (PerkinElmer) was used for building the spectral libraries and unmixing MSIs, as well as for subsequent analysis including cell segmentation and phenotyping.

**Staining with SNA and MALII on normal and PDAC tissue.** FFPE tissue was deparaffinised and antigen retrieval was performed using Tris-EDTA buffer (pH 9) and microwave treatment. Tissue was then blocked with Carbo-Free Blocking Solution (Vector Labs) and incubated with biotinylated Maackia Amurensis Lectin II (MALII) for 30 min at room temperature. Slides were washed and incubated with Streptavidine–Peroxidase conjugates for 30 min at room temperature. Development of section was performed with 3,3'-Diaminobenzidine (DAB, Abcam) and using haematoxylin as counterstain. ImageJ was used for quantification of the signal intensity of the ductal cells using the *colour deconvolution plug-in*. Tissue analysis and duct selection were corroborated by pathologist.

**Phosphorylation analysis.** The phosphorylation of Siglec receptors induced by the Sialic acid dendrimers was analysed using the Human Phospho-Immunoreceptor Array Kit (R&D Systems) according to the manufacturer's instructions. For the generation of cell lysates, CD14+ isolated monocytes were first incubated in FCS-free RPMI for 4 h at 4 °C and then stimulated with 5 μM of sialic acid- or control dendrimers for 30 min. After this, supernatant was discarded and cells were lysed using the lysis buffer from the kit.

**Characterisation of sialylated glycans on PDAC cells**
*Flow Cytometry.* All the staining with plant lectins (Vector Laboratories) or Chimeric receptors (R&D System) were performed in 1% BSA in HBSS supplemented with calcium and magnesium (Gibco). For the analysis using plant lectins or in-house Fc chimeric proteins, 100.000 cells were incubated with 1 μg/mL of biotinylated plant lectin or 1 μg/mL Fc chimeras for 45 min at 4 °C. Siglec-Fc chimeras were pre-incubated FITC-conjugated anti-human IgG (Jackson ImmunoResearch) for 15 min at room temperature before adding to the cells. The detection of plant lectins was performed by incubating the cells with AlexaFluor 488-conjugated Streptavidin (Invitrogen).

When indicated, cells were treated with 30mU/mL of Neuraminidase from *Arthrobacter ureafaciens* for 30 min at 37 °C to study the sialic acid dependency of the interaction. To evaluate whether sialylated structures are present in *N*-glycans,

*O*-glycans or Glycolipids, cells were treated for 3 days with 2 mM Benzyl-GalNAc, 10 μg/mL Kifunensine or 1 mM PPMP, respectively.

*siRNA.* To study the role of different 2,3 sialyltransferases in the synthesis of Sigle-7 and Siglec-9 ligands, we knocked-down their expression by using SMARTpools (Dharmacon), consisting of 4 different small interfering RNA (siRNA) that target a given gene. Cell lines were transfected with SMARTpools targeting the genes ST3GAL1-5 using DharmaFECT 2 Transfection Reagent (Dharmacon). After 3 days, cells were analysed for expression of Siglec ligands by flow cytometry. As control, a non-targeting siRNA pool was used (Dharmacon).

*ELISA-based assay for sialoglycans.* Lysates were obtained by extensive washing with ice cold PBS and incubation in the culture flask with RIPA buffer (50 mM Tris-HCl pH = 8, 150 mM NaCl, 2 mM EDTA, 1% NP-40, 0.5% Sodium Deoxycholate and 0.1% SDS) supplemented with protease inhibitors (Roche). Trypsinization of the cells was avoided to preserve epitopes. Protein concentration was performed by the bicinchoninic acid (BCA) assay (Pierce). To study whether the cell lines were able to secrete sialylated-conjugates capable to interact with Siglec-7 and Siglec-9, we performed an ELISA in FCS-free supernatant of the cell lines. 2 × 10^6 cells were incubated overnight in a T75 flask in RPMI 1640 (Gibco) supplemented with 10% Fetal Calf Serum (Lonza), 2 mM L-Glutamine (Gibco) and 1000 U/mL Penicillin-Streptomycin (Gibco). Cells were extensively washed with PBS and incubated for 48 h with RPMI 1640 (Gibco) supplemented with 2 mM L-Glutamine (Gibco) and 1000 U/mL Penicillin-Streptomycin (Gibco). Supernatants were harvested and lyophilised. Protein concentration was determined by BCA assay (Pierce).

Different concentrations of FCS-free supernatant (25 μg to 0.5 μg) or 2 μg of total cell lysates were coated in NUNC maxisorp plates overnight at room temperature. Plate was washed twice with PBS, blocked with Carbo Free Blocking Buffer (Vector Laboratories) and incubated with 0.2 μg/mL of Siglec9-hFc or 20 μg/mL of Siglec7-hFc. Detection was performed with HRP-conjugated anti-human IgG Fc (Jackson ImmunoResearch) and revealed using TMB.

**CRISPR-Cas9 gene knockout.** The generation of CMAS knockout of the cell line BxPC3 was performed as reported before. Briefly, sgRNA strands for human CMAS gene were phosphorylated and annealed prior to cloning in the pSpCas9 (BB)-2A-Puro plasmid, a gift from Feng Zhang (Addgene#62988). The following gRNA strands for human *CMAS* were used: top strand CACCGATATCTGAACAGTGTAT; bottom strand AAACATACACTGTTCAGATATC. BxPC3 cells were transfected using Lipofectamine™ LTX with PLUS™ Reagent (Invitrogen), selected with puromycin and sorted based on negative recognition of the plant lectin MAL-II using BD FACSAria™ Fusion FACS sorter. Mock cell lines were transfected with the pSpCas9(BB)-2A-Puro plasmid without guide RNA.

**Transcriptomics analysis.** Characteristics of the data sets used in the different analysis of this paper are described in the Supplementary Table 1.

*Bulk sequencing.* The package *limma* was used for the analysis of the differential gene expression, with FDR correction for multiple comparisons. To identify clinically relevant glyco-code-related genes in pancreatic cancer, analysis between normal and tumour tissue was performed in paired (GSE15471 and GSE62452) or unpaired (GSE16515 and GSE71729) samples.

Gene set enrichment analysis in a single-sample basis using the *GSVA* package[54]. For survival analysis, patients were ordered based on a given score and the top and bottom thirds were defined as high and low expression, respectively (Supplementary Fig. 4C). Log-rank test was performed to determine statistical significance in survival analysis, as implemented in the function *ggsurvplot* of the package *survminer*.

*Single-cell RNA-seq analysis.* The scRNA-seq data previously published by Peng et al. was downloaded from the Genome Sequence Archive project PRJCA001063 as pre-processed row data and imported into the package *Seurat* (v3) for downstream analysis (Supplemenary Table 1)[24,55]. The *SCTransform* function was used to normalise and scale the data, regressing out the mitochondrial percentage, and Principal Component Analysis (PCA) was performed with the 3000 most variable genes. PCA components 1 to 10 were used for graphical based clustering at a resolution of 1. Clusters were grouped based on the superposition with the clusters previously identified by Peng et al.[24]. These clusters were projected onto Uniform Manifold Approximation and Projection (UMAP) dimensional reduction.

For the characterisation of the myeloid cells present in PDAC, we selected the myeloid cluster identified previously, re-normalised using the *SCTransform* function and clustered using the first 8 components of a new PCA, at a resolution of 0.2.

Gene sets specific for each myeloid population were generated using the function *FindMarkersAll* in the *Seurat* package to find differential genes expressed between each cluster against: (i) other myeloid populations; (ii) other populations present in PDAC. Common genes between both conditions were kept, generating gene sets that were later used in survival analysis using bulk transcriptomics.

Differentiation analysis of monocytes towards moMac and moDC was done using diffusion maps as implemented in the R package *destiny*[56]. For the calculation of the *pseudotime*, we used the package *slingshot*[57].

*NicheNet* was used to infer tumour-derived factors that mediate the monocyte to macrophage differentiation[30]. *NicheNet* can use expression data of effector cells and target cells to predict ligand-receptor pairs able to induce given phenotype changes in the latter, based on models build on the current knowledge of signalling and gene regulation networks. For our analysis, we studied genes expressed in the sender cells (in our case, tumour cells) capable to interact with receptors present in the 'receiver' cells (monocytes) and induce the expression of a gene set associated with the moMac cluster. To generate the said gene set, we performed differential gene expression between moMac and Monocytes using the *Seurat* function *FindMarkers* and selecting the genes enriched in moMac (logFC ≥ 0.75 or the difference in the percentage of expressing cells between moMac and Monocytes is equal or higher to 25%). Only extablished ligand–receptors pair presents in curated databases were used.

**Statistical analysis**. The analysis of single-cell transcriptomic data was performed in R (v 3.5.1). GraphPad Prism was used for the rest of the analyses. Specific test performed is indicated in each figure. Log-rank test was performed to determine statistical significance in survival analysis, as implemented in the function *ggsurvplot* of the package *survminer*. All the graphs are represented as the mean ± standard error of the mean (SEM). Relative expression in flow cytometry (Figs. 2C, 2D, 5F, 6B and 6H) was calculated as follows: first, ΔMFI was calculated for each sample as $gMFI_{Sample} - gMFI_{FMO}$ (as obtained from the softare FlowJo), to later divide each ΔMFI over the $ΔMFI_{Control}$. Controls are indicated in the respective figure. Similarly, the relative expression of Siglecs ligands determined by ELISA (Fig. 2B) was calculated as the O.D. at 450 nm of the sample over one of the positive controls.

**Reporting summary**. Further information on research design is available in the Nature Research Reporting Summary linked to this article.

## Data availability
The transcriptomic data that support the findings of this study are publicly available, as detailed in the Supplementary Table 1. For the analysis of differential gene expression between normal and tumour tissue was obtained from the NCBI GEO database (https://www.ncbi.nlm.nih.gov/geo/) with the following accession numbers: GSE15471[17], GSE16515[18], GSE62452[19], GSE71729[20]. The single-cell RNA-Seq data from Peng et al. was downloaded from the Genome Sequence Archive (https://bigd.big.ac.cn/gsa/) under the project PRJCA001063[58]. The data from the PAAD project of the TCGA is available for download from the Broad Institute GDAC Firehose (https://gdac.broadinstitut.org)[28]. The transcriptomic data published by Puleo et al. was downloaded from the ArrayExpress database with the accession number E-MTAB-6134[26]. Data from PACA-AU project of the ICGC (release 25) was obtained from https://dcc.icgc.org/ [27]. The remaining data are available within the article, Supplementary Information or available from the authors upon request.

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

## Acknowledgements

The authors would like to acknowledge the Microscopy and Cytometry Core Facility at the Amsterdam UMC-Location VUmc for providing assistance in cytometry experiments. This work is financially supported by Immunoshape (MSCA-ITN-2014-ETN No. 642870) to E.R.; by SPINOZA prize to E.R. and Y.K.; by the European Research Council (ERC-339977-Glycotreat) to S.T.T.S. and Y.K.; by KWF VU2014-7200 to K.Bo.; by the AIRC Start-Up grant (to E.G.).

## Author contributions

E.R. and Y.K. conceived the study. E.R., K.Bo., E.L., L.K., S.B., T.E., S.S., M.C., J.H., G.K., S.V., E.G., J.J.G.V. and Y.K. designed experiments or provided essential technical support. E.R., K.Bo., K.Br., S.B. acquired experimental data. E.R. K.Bo. and K.Br. analysed data. E.R. performed transcriptomic analyses. N.G. collaborated with the pathology assessment of tumour tissues. K.Bo., E.R. and Y.K. drafted the manuscript. E.R., K.Bo. K.Br., G.K., S.V., E.G., J.J.G.V. and Y.K. provided critical intellectual content. Y.K. supervised the study.

## Competing interests

The authors declare no competing interests.
