## [Peer Review File · Nature Communications]

REVIEWER COMMENTS

Reviewer #1 (Remarks to the Author):

In this paper, Rodriguez and colleagues focus on the role of sialic acids and siglecs in the interaction of tumor cells with immune cells. They claim that pancreatic tumors display increased levels of sialylated glycan structures, which can be recognised by the lectin receptors Siglec-7 and -9 on myeloid cells.

They propose that sialylated glycans drive the differentiation of monocytes to macrophages, which then potentially repress T-cells and NK cells.

This paper contains lots of information and represents a significant amount of work. The message, that glycans structure are use by tumors to regulate immune cells and orient them towards being tumor tolerant, is interesting. However, several issues moderate my enthusiasm. Among them, most of the signals seem to be of moderate intensity, given the impression that the authors are stretching somewhat the interpretation of the data. A more balanced reporting of the data would improve the scientific quality of the manuscript.

There is a lack of quantitative statements in general, leading to qualitative, over-reaching statements.

Specific points:

For instance, in the first paragraph (Fig 1); the authors claim that enzymes involved in alpha 2,3 sialylation are upregulated. However, based on their graph in Fig 1B, this up-regulation appears modest (1.2 fold?), with a limited p-value.

The enzyme that is most clearly up-regulated is ST6GALNAC1, but it is apparently not involved in the Siglec biology and is not studied here.

2. More problematic is the lack of quantitation for MAL-II in their tumor samples. The authors only show one example; how much is the increase and in what fraction of the tumors do they observe this phenotype?

The cell line data is not that informative since there is no comparison with "normal pancreatic cells".

3. Figure 2: The authors claim that:

Line 166: "... all myeloid populations, including monocytes, showed expression of both Siglec-7 and Siglec-9. However, according to their graphs in Fig2F and 3C, the proportion of cells expressing these lectins is between 10 and 20%. This important fact is not mentioned nor discussed as far as I could see.

Also, the transcriptomic data is not consistent with the FACS data presented in 5B.

4. The fact that a minority of cells express the Siglec makes it hard to rationalise the large effects of sialic acids observed in Fig 6A, C and I. A knock-down of each Siglec would be required to make sure that the effects observed are indeed due to these cell surfaces receptors.

Minor issues:

Lane 96: Ref missing

Reviewer #2 (Remarks to the Author):

Rodriguez et al have evaluated sialic acid in PDAC cell lines and tumor tissues. They identify Siglec-7 and Siglec-9 ligands specifically in PDAC cell lines. Knockdown of several sialytransferases with siRNA and small molecule inhibitors identify ST3GAL1 and ST3GAL4 as being important for these Siglec ligands. Siglec expression was primarily in myeloid cells in PDAC. scRNA-seq analysis

noted SIGLECs in Monocytes/macrophages lineages with moMac and moDCs diverging. They then do co-culture functional experiments with monocytes and cancer cell lines. They then use dendrimers to carry sialylated structures to verify Siglec ligands are likely driving moMAC differentiation. Overall the paper is much better focused than the previous version with interesting functional data. Attention to figure legends is needed as many are not appropriately describing the figure or over-interpreting the data in the figure.

Comments

1. Figure 2 E

a. Did the authors just pool STGAL4+STGAL1 expression for survival analysis and how did they make a call on HIGH vs LOW? Was this the median? How many patients in each group? Probably would be best to show STGAL4 and STGAL1 alone instead of pool

b. small grammar and spelling based "on" the ... "Log-rank" test

2. Figure 3

a. E – Spearman correlation ... should be here instead of F

b. F – Differentiation analysis of moMac and moDC is not really informative. Should explain what is "pseudotime". Is this a UMAP?

c. G – is this analysis of single cell data across how all of these patients? I assume not. Again how many patients in each group and what is the cutoff?

3. Figure 4 The authors should avoid over-interpretation in the legend.

a. B) "These data suggest that tumor cells may directly contribute to the differentiation of moMAC". Probably would be better as "proposed model of tumor cell monocyte differentiation.

b. E) PDAC tumor cell lines differentiate monocyte into macrophages. Maybe should say % CD163 cells of CD14 cells when co-cultured with cell line (x-axis).

c. F) Not very specific of what I'm looking at. 3 dimensional UMAP?

4. Figure 5- same comment as Figure 4. There are general statements of "analyzed by flow cytometry, but they should tell us what are we looking at? Is the y axis FITC/Cy3,. What is the X-axis.

Reviewer #3 (Remarks to the Author):

This study describes how differences in glycosylation can affect macrophage activation and polarisation in pancreatic cancer. The manuscript is a mixture of bioinformatic analysis and the use of experimental data. The authors identify that the amount of sialylated glycans are increased in human PDAC tissue compared to normal pancreas and that sialylated glycans can drive macrophage polarisation towards an immune-suppressive, M2-like phenotype through their stimulation of the lectin receptors Siglec 7 and 9 on macrophages. These are interesting observations in regard to pancreatic cancer, but similar findings have been reported for other cancer types. For example that sialic acids regulate the activation of macrophages is well published (PMID 17380156) and that aberrant glycosylation in cancer can drive myeloid cell polarisation towards an immunosuppressive phenotype via Siglec 9 has already been reported by other groups (PMID 27595232; PMID 25225409).

In addition, the characterisation of macrophage activation and polarisation in response to sialic acid is based on the use of PDAC cells lines and 2D cell culture assays and therefore the interpretation of the observed results is somewhat limited.

Additional points:

Fig 1D: Was the MAL II lectin staining only performed on one PDAC tissue section? Quantification and scale bars are missing. More important: tissue staining shows that increased sialylation is not specific for cancer cells, but there is also a marked increase in sialylation in the surrounding tissue. Is the claimed macrophage polarisation only mediated by sialic acids derived from cancer cells or can sialic acids derived from other cell types also contribute to macrophage polarisation?

Supplement Fig S1: please add a positive control staining to the panel to show that your cell lines are truly negative.

Fig 2A-C: results are based on a panel of 6-9 established PDAC cell lines. What about primary isolated PDAC cells? Do they also mainly express ligands for Siglec 7 and 9?

Fig 3A: Please provide further information about how the different clusters of monocytes observed in the scRNA Seq data set were annotated and defined (MoMac; MoDc etc)? Were the functions annotated based on the expression of the selected markers listed by the authors in the main text or what type of analysis was performed?

Fig 3C: the percentage of Siglec 7/9 expression in monocytes seems to be rather low (around 10-20 % of each monocyte population). Thus, please comment on how does sialic acid polarise the remaining 80-90 % of cells which do not express Siglec 7 or 9?

Fig 4: it has been shown for many cancer types, including pancreatic cancer, that tumour cells secrete CSF-1 and that CSF-1 drives monocyte to macrophage differentiation. Thus, the authors findings confirm previous published data. However, the claim that reduced cancer cell numbers (reduced cancer cellularity) drives dendritic cell differentiation is not supported by any experimental data. Several reports have shown that in PDAC, high numbers of DCs is critical for an effective anti-tumour immune response, resulting in smaller tumour burden (PMID: 32183949). Thus, the statement that fewer tumour cells lead to more DCs might rather be a consequence than a cause.

Fig 5: Please explain how "relative expression" of CD206 and CD163 was assessed in panel F.

Fig 6B: Please explain how "relative expression" was assessed.

Fig 6G-I: Please explain the role of LPS in this assay and discuss the relevance of LPS during macrophage polarisation in the tumour microenvironment.

Fig 7: Schematic only, no data shown. Thus, this figure should be combined with figure 6 or be move into supplement.

Reviewer #4 (Remarks to the Author):

All raised aspects has been addressed. The paper 2 in current form is significantly improved and clear.

REVIEWER COMMENTS

Reviewer #1 (Remarks to the Author):

In this paper, Rodriguez and colleagues focus on the role of sialic acids and siglecs in the interaction of tumour cells with immune cells. They claim that pancreatic tumours display increased levels of sialylated glycan structures, which can be recognised by the lectin receptors Siglec-7 and -9 on myeloid cells. They propose that sialylated glycans drive the differentiation of monocytes to macrophages, which then potentially repress T-cells and NK cells.

This paper contains lots of information and represents a significant amount of work. The message, that glycans structure are use by tumours to regulate immune cells and orient them towards being tumour tolerant, is interesting. However, several issues moderate my enthusiasm. Among them, most of the signals seem to be of moderate intensity, given the impression that the authors are stretching somewhat the interpretation of the data. A more balanced reporting of the data would improve the scientific quality of the manuscript.

There is a lack of quantitative statements in general, leading to qualitative, over-reaching statements.

We thank the reviewer for the analysis of the manuscript and the comments provided. We have changed the manuscript according to the comments of the reviewer to include more quantitative data and statements gauged in the quantitative data provided.

We would like to elaborate on the confusion around the number of myeloid cells expressing Siglec receptors and the difference between the transcriptomic data and the flow cytometry (FACS) data. In single-cell RNA sequencing, the number of genes detected in each cell depends on the sequencing depth. As example, with 50,000 read pairs/cell for RNA-rich cells such as cell lines, only 30-50% sequencing saturation may be reached. Therefore, it is likely that not all the Siglec RNA's in each cell are detected¹.

Here we quote from the review of Haque *et al* on single cell RNA sequencing¹: *“The efficiency with which poly-adenylated mRNA species are captured, converted into cDNA and amplified is currently unclear, and, depending on the study, can range between 10 and 40% [13, 44, 64, 65]. This means that, even if a gene is being expressed, perhaps at a low level, there is a certain probability that it will not be detected by current scRNA-seq methods.”*

In the data set of Peng *et al*, a median of 2205 genes per cell are detected, with a range between 222 and 7973². This means that the percentage of cells where transcripts of the Siglec receptors are detected (Figure 2F and 3C) is not appropriate for the quantitative assessment of their expression, but as a qualitative tool. Therefore, alternative methods are necessary for a better quantitative measure of the Siglec receptors, as is flow cytometry. In Figure 5B we confirmed that monocytes indeed express Siglec-7 and Siglec-9 using FACS. The following figure shows that even the transcripts of genes corresponding to markers of different populations are not found in all cells. In this case, the transcripts of *CD14*, *CD19* and *CD3G* can be found in ~50% of Myeloid, B cells and T cells, respectively.

Specific points made by reviewer 1:

1. For instance, in the first paragraph (Fig 1); the authors claim that enzymes involved in alpha 2,3 sialylation are upregulated. However, based on their graph in Fig 1B, this up-regulation appears modest (1.2 fold?), with a limited p-value.

2. More problematic is the lack of quantitation for MAL-II in their tumour samples. The authors only show one example; how much is the increase and in what fraction of the tumours do they observe this phenotype? The cell line data is not that informative since there is no comparison with “normal pancreatic cells”.

We agree with the reviewer that our results were not correctly presented in Figure 1. We have now made several changes to the figure and the manuscript to include several points:

- In the first section of the results, we have changed the focus and now start with general upregulation of sialic acid, instead of emphasizing immediately on the increase of α 2,3-linked sialic acids.
- We changed the transcriptomic analysis of the different sialylation pathways depicted in Figure 1A, using now gene set enrichment analysis. This is a better measure to assess if a given pathway, as a whole, is up- or downregulated in tumour tissue, considering all its genes. This shows a general upregulation of sialylation in PDAC and in particular the pathways associated with synthesis of the sugar donor and enzymes that catalyze the addition of α 2,3- and α 2,6-linked sialic acids. As a supplementary figure, we also included the differential gene expression of sialic acid-related genes. Now, this analysis includes four data sets and do not separate between the different glycosylation clusters described in the other manuscript, as asked by the editor.
- Also, we included the analysis of sialic acids in a “normal” cell line and in primary cell lines by ELISA in Figure 1, which shows the presence of α 2,6-linked sialic acids in the normal duct cell line but absence of α 2,3-linked sialic acids.
- We performed and quantified immunohistochemistry to analyze the presence of α 2,3-linked and α 2,6-linked sialic acids in tumour tissue. In the Supplementary Figure 1, we show examples of the immunohistochemistry stainings and the strategy for the specific quantification of

ductal cells. Moreover, we changed the selection of images to better display the clear differences found.

- The broad presence of α 2,3-linked sialic acids in PDAC cell lines, it's absence on normal-like pancreatic ductal cell line (HPNE) compared to α 2,6-linked sialic acids, and the potential to serve as candidates for Siglec ligands in PDAC cell lines, led us to explore this in this manuscript. This motivated us to study the ST3GAL genes in Figure 2.

The enzyme that is most clearly up-regulated is ST6GALNAC1, but it is apparently not involved in the Siglec biology and is not studied here.

ST6GALNAC1 was indeed found to be upregulated in the RNA sequencing datasets. ST6GALNAC1 is the transcript encoding for the enzyme involved in the synthesis of the sialyl-Tn antigen. We also evaluated the sialyl-Tn expression on the PDAC tumour cell lines, which was expressed at moderate levels. Given the high expression of Siglec-7/9 ligands in these cell lines, seems unlikely that sialyl-Tn is a main contributor.

3. Figure 2: The authors claim that:

Line 166: "... all myeloid populations, including monocytes, showed expression of both Siglec-7 and Siglec-9. However, according to their graphs in Fig2F and 3C, the proportion of cells expressing these lectins is between 10 and 20%. This important fact is not mentioned nor discussed as far as I could see.

Also, the transcriptomic data is not consistent with the FACS data presented in 5B.

We hope we have convinced you that the percentage of expressing cells in single cell sequencing is not a good quantitative measurement and why does not match with the Figure 5B. However, we included in the revised manuscript a phrase to highlight this issue (Page 7, line 157-159).

4. The fact that a minority of cells express the Siglec makes it hard to rationalise the large effects of sialic acids observed in Fig 6A, C and I.

Based on our explanation about the difference between FACS data and single cell transcriptomics data described in the beginning of our rebuttal, and in combination with literature on Siglec expression, we hope we have now convinced you that it is not a minority of myeloid cells that express Siglec, we explained this also clearly in the manuscript.

A knock-down of each Siglec would be required to make sure that the effects observed are indeed due to these cell surfaces receptors.

Generating a knock down of Siglec receptors is technically very challenging with primary cells. The transfection of myeloid cells often leads to considerable cell death and cell activation. Thus, in particular when analyzing differentiation processes and activation processes it is questionable at all whether this would provide concluding results. This is one of the reasons why we choose using blocking antibodies for Siglec-7 and Siglec-9 (separate and in combination) in primary 'untouched' human cells,

which indeed provides the evidence that the effect on monocyte differentiation is mediated via Siglec-7 and Siglec-9 (Figure 5C).

Minor issues:

Lane 96: Ref missing

We included a reference

Reviewer #2 (Remarks to the Author):

Rodriguez et al have evaluated sialic acid in PDAC cell lines and tumour tissues. They identify Siglec-7 and Siglec-9 ligands specifically in PDAC cell lines. Knockdown of several sialytransferases with siRNA and small molecule inhibitors identify ST3GAL1 and ST3GAL4 as being important for these Siglec ligands. Siglec expression was primarily in myeloid cells in PDAC. scRNA-seq analysis noted SIGLECs in Monocytes/macrophages lineages with moMac and moDCs diverging. They then do co-culture functional experiments with monocytes and cancer cell lines. They then use dendrimers to carry sialylated structures to verify Siglec ligands are likely driving moMAC differentiation. Overall the paper is much better focused than the previous version with interesting functional data. Attention to figure legends is needed as many are not appropriately describing the figure or over-interpreting the data in the figure.

Comments

1. Figure 2 E

a. Did the authors just pool STGAL4+STGAL1 expression for survival analysis and how did they make a call on HIGH vs LOW? Was this the median? How many patients in each group? Probably would be best to show STGAL4 and STGAL1 alone instead of pool

Thank you for your remarks and suggestions. We indeed used the mean expression of pooled ST3GAL4 + ST3GAL1 for survival analysis. For defining the High vs Low patients, we used the top and bottom 1/3 of the patients. We have now changed Figure 2E to include the number of patients and also specified more details in the legend of the figure and in the method section (Page 21, lines 548-551).

We do not see differences in the survival when use ST3GAL1 and ST3GAL4 separated. This may be because the expression of one may compensate for the absence of the other, given that the triggering of both Siglec-7 and Siglec-9 can contribute to the differentiation of TAMs (Page 6, lines 146-147).

b. small grammar and spelling based “on” the ... “Log-rank” test

We have adjusted it.

2. Figure 3

a. E – Spearman correlation ... should be here instead of F

We have adjusted it.

b. F – Differentiation analysis of moMac and moDC is not really informative. Should explain what is “pseudotime”. Is this a UMAP?

We disagree with this comment. We suggest in our manuscript that monocytes that infiltrate the tumour can differentiate into macrophages or dendritic cells. The results of this analysis suggest that this transition of monocyte differentiation processes towards moMac or moDC can be detected in PDAC tumours.

Diffusion maps is a dimensional reduction tool widely used to analyze trajectories during cell differentiation. In this algorithm, individual points represent cells along a continuum undergoing

gradual transcriptional changes. This differentiation is reflected by an underlying temporal variable called *pseudotime*, that can be seen as measure of the progression through the differentiation process. There are different algorithms that can be used to infer pseudotime from single cell transcriptomics or proteomics. This analysis is widely used in various studies³⁻⁵.

c. G – is this analysis of single cell data across how all of these patients? I assume not. Again how many patients in each group and what is the cutoff?

The survival analysis of Figure 3G was done using a bulk RNA-sequencing dataset. The workflow used is now better explained in the Supplementary Figure S4C. First, we analyzed the single cell transcriptomic data to define a series of genes characteristics of each of the myeloid cluster, and then used those to generate gene sets to be used in gene set enrichment analysis in bulk transcriptomics.

We have changed this in the manuscript and the figure, to explain better the procedure that we followed for this analysis (Bulk transcriptomics: Page 21, lines 548-551; Gene set: Page 22, lines 564-567).

3. Figure 4 The authors should avoid over-interpretation in the legend.

a. B) “These data suggest that tumour cells may directly contribute to the differentiation of moMAC”. Probably would be better as “proposed model of tumour cell monocyte differentiation.

We changed the figure legends in the manuscript.

b. E) PDAC tumour cell lines differentiate monocyte into macrophages. Maybe should say % CD163 cells of CD14 cells when co-cultured with cell line (x-axis).

We have changed it.

c. F) Not very specific of what I’m looking at. 3 dimensional UMAP?

The graph presented in the Figure 4F is a 3-dimensional diffusion map that we used to study the monocyte differentiation *in vitro*. We choose to present the data as a 3-dimensional graph (using the first 3 variables generated in by the algorithm of diffusion map – DM1-3), to more clearly show the differences between the different condition, especially when compared with *in vitro* generated monocyte derived dendritic cells (moDCs). Given that cell lines differentiate monocytes into two different populations (namely TAMS and CD163^{lo} Myeloid cells), two dimensions are not enough to observe correctly observe the 4 different cell types (Monocytes, moDCs, TAMS and CD163^{lo} cells) and may lead to wrong conclusions. For example, the use of a 2-dimensional graph visualization of the monocyte differentiation by cell lines (using only DM1 and DM2), would not allow to correctly visualize the correct position of moDCs. For clarity, we added the variables names to the graph that were missing in the previous submission.

We provided now more details in the text about this analysis (Page 8, line 196).

4. Figure 5- same comment as Figure 4. There are general statements of “analyzed by flow cytometry, but they should tell us what are we looking at? Is the y axis FITC/Cy3,. What is the X-axis.

We modified the Figure and the legends to include the missing information.

Reviewer #3 (Remarks to the Author):

This study describes how differences in glycosylation can affect macrophage activation and polarisation in pancreatic cancer. The manuscript is a mixture of bioinformatic analysis and the use of experimental data. The authors identify that the amount of sialylated glycans are increased in human PDAC tissue compared to normal pancreas and that sialylated glycans can drive macrophage polarisation towards an immune-suppressive, M2-like phenotype through their stimulation of the lectin receptors Siglec 7 and 9 on macrophages. These are interesting observations in regard to pancreatic cancer, but similar findings have been reported for other cancer types. For example that sialic acids regulate the activation of macrophages is well published (PMID 17380156) and that aberrant glycosylation in cancer can drive myeloid cell polarisation towards an immunosuppressive phenotype via Siglec 9 has already been reported by other groups (PMID 27595232; PMID 25225409). In addition, the characterisation of macrophage activation and polarisation in response to sialic acid is based on the use of PDAC cells lines and 2D cell culture assays and therefore the interpretation of the observed results is somewhat limited.

We agree with the reviewer that the interaction of sialylated glycans with myeloid cells via Siglec receptors have been reported in literature. However we believe that our study provides a novel clinical context for the immunomodulatory effects of sialic acids, as is pancreatic cancer, including also new molecular details and new bioinformatics analysis to explore glycosylation, from patients cohort of primary tissue to tumour cell lines. Our study highlights that using bioinformatics analysis, provide us novel information on specific sialyltransferases expression predict survival benefit with immune modulation via Siglec-7 and Siglec-9. Moreover, our study shows that using single cell RNA-seq you can link this information to myeloid subset differentiation and Siglec expression in tumour infiltrating myeloid cells. We believe that connecting bioinformatics analysis using patients cohorts, and validation in primary tumour-tissues and tumour cell-line immune cell co-cultures, using CRISPR-Cas9, siRNA, our paper set-stage not only for PDAC but also for new discoveries potential of tumour glycosylation-immune relation not only in PDAC but also other types of cancer. In this respect we disagree with the reviewer that results are somewhat limited.

Additional points:

Fig 1D: Was the MAL II lectin staining only performed on one PDAC tissue section? Quantification and scale bars are missing. More important: tissue staining shows that increased sialylation is not specific for cancer cells, but there is also a marked increase in sialylation in the surrounding tissue. Is the claimed macrophage polarisation only mediated by sialic acids derived from cancer cells or can sialic acids derived from other cell types also contribute to macrophage polarisation?

We have included quantification of SNA and MAL-II lectin staining in several PDAC tumour tissue as well as healthy pancreas tissue. Several of the tumour biopsies also contained histologically normal tissue (Figure 1, Figure S1).

Indeed, sialic acid are also found in tumour microenvironment as well. In Figure 5A we show that Siglec-7/9 ligands are also secreted by PDAC tumour cells. This can explain the sialic acid expression found in the tumour microenvironment. We cannot exclude the possibility that other cells in the tumour microenvironment also contribute to sialic acid mediated macrophage polarization, however this is a separate topic and study on itself. We included this part in the discussion (Page 12, lines 309-315).

Supplement Fig S1: please add a positive control staining to the panel to show that your cell lines are truly negative.

We were unable to find cell lines that serve as positive control for the FACS stainings using the Siglec-3 and Siglec-5 fc constructs. However, we do see that these constructs can recognize some sialylated structures in an ELISA assay, showing that the constructs are functional.

Positive Controls used:

Siglec-3 hFc – COVID-Spike Protein; Siglec-5 hFc – PAA α 2,3 Sialic Acid; Siglec-7 hFc – PAA α 2,6 Sialic Acid; Siglec-9 hFc – PAA α 2,3 Sialic Acid; Siglec-10 hFc – PAA α 2,3 Sialic Acid.

Fig 2A-C: results are based on a panel of 6-9 established PDAC cell lines. What about primary isolated PDAC cells? Do they also mainly express ligands for Siglec 7 and 9?

In this new submission, we have included the analysis of primary PDAC cell lines. Using the ELISA mentioned before, we indeed see that primary cell lines express Siglec-7 and Siglec-9 (Figure 2B).

Fig 3A: Please provide further information about how the different clusters of monocytes observed in the scRNA Seq data set were annotated and defined (MoMac; MoDc etc)? Were the functions annotated based on the expression of the selected markers listed by the authors in the main text or what type of analysis was performed?

The annotation of the different cluster was based in two different analysis: the differential gene expression (DGE) between the clusters and gene set enrichment analysis.

For the differential gene expression, we used the function *FindAllMarkers* of the Seurat package. This analysis performs differential gene expression between a given cluster against all the others. Each cluster was assigned to a certain myeloid population given the expression of broadly used myeloid markers. For example, moMac were annotated as such given the expression of *MARCO*, *CD68* and *CCL2*, among others. A table with the results of the DGE analysis has been included as supplementary Table 2.

For the enrichment analysis we use the gene sets of different *in vitro* generated myeloid cells published by Sander *et al*, using the function *AddModuleScore* of the package *Seurat* (Figure S3A) ⁶. The gene sets include gene associated to Monocytes, moDCs and moMac generated with both M-CSF and GM-CSF (a

list with the genes used can be found in the supplementary Table 3). This analysis result in a series of Scores associated to each of these gene sets, which were plot in the UMAP graph (as shown in Figure S3A). This data clearly shows an association of the different scores with the respective clusters identified in PDAC. For a clearer representation of the data, we have now changed the figure to also included violin plot of each Scores.

We have now provided a better explanation to the annotation of each cluster (Page 7, line 173-188).

Fig 3C: the percentage of Siglec 7/9 expression in monocytes seems to be rather low (around 10-20 % of each monocyte population). Thus, please comment on how does sialic acid polarise the remaining 80-90 % of cells which do not express Siglec 7 or 9?

In single-cell RNA sequencing, the number of genes detected in each cell depends on the sequencing depth. As example, with 50,000 read pairs/cell for RNA-rich cells such as cell lines, only 30-50% sequencing saturation may be reached. Therefore, it is likely that not all the Siglec RNA's in each cell are detected¹.

Here we quote from the review of Haque *et al* on single cell RNA sequencing¹: “The efficiency with which poly-adenylated mRNA species are captured, converted into cDNA and amplified is currently unclear, and, depending on the study, can range between 10 and 40% [13, 44, 64, 65]. This means that, even if a gene is being expressed, perhaps at a low level, there is a certain probability that it will not be detected by current scRNA-seq methods.”

In the data set of Peng *et al*, a median of 2205 genes per cell are detected, with a range between 222 and 7973². This means that the percentage of cells where transcripts of the Siglec receptors are detected (Figure 2F and 3C) is not appropriate for the quantitative assessment of their expression, but as a qualitative tool. Therefore, alternative methods are necessary for a better quantitative measure of the Siglec receptors, as is flow cytometry. In Figure 5B we confirmed that monocytes indeed express Siglec-7 and Siglec-9 using FACS. The following figure shows that even the transcripts of genes corresponding to markers of different populations are not found in all cells. In this case, the transcripts of *CD14*, *CD19* and *CD3G* can be found in ~50% of Myeloid, B cells and T cells, respectively. We have added a comment in the revised manuscript (page 7, line 157-159).

Fig 4: It has been shown for many cancer types, including pancreatic cancer, that tumour cells secrete CSF-1 and that CSF-1 drives monocyte to macrophage differentiation. Thus, the authors' findings confirm previous published data. However, the claim that reduced cancer cell numbers (reduced cancer cellularity) drives dendritic cell differentiation is not supported by any experimental data. Several reports have shown that in PDAC, high numbers of DCs is critical for an effective anti-tumour immune response, resulting in smaller tumour burden (PMID: 32183949). Thus, the statement that fewer tumour cells lead to more DCs might rather be a consequence than a cause.

We agree with the reviewer. We adjusted the text to be more carefully that we see a correlation between cancer cell numbers and moMac numbers, discussing the potential interactions that may exist between the tumour and the myeloid cells.

It must be noted that the definition of moMac or moDC enriched tumours is taking into account only the myeloid population, and not the total amount of cells of the tumour. This means that we are not analyzing the absolute number of a given type of cell. This is to identify factors that may influence myeloid differentiation independent of the total infiltration of monocytes to the tumour. To clarify this we changed the terms "moDC/moMac- enriched tumours" that may lead to confusion. We also adjusted this in the text

Fig 5: Please explain how "relative expression" of CD206 and CD163 was assessed in panel F.

Geometric mean intensity (gMFI) of CD206 and CD163 of the different condition was obtained by flowjo software. Δ MFI was calculated by subtracting the gMFI of the FMO to the gMFI of the sample. Since the samples were paired, we calculated the relative expression by dividing the Δ MFI of the CMAS KO cells over the Δ MFI of the control condition. We clarify this in the methods section and in the legend of the figure (Page 22, line 587).

Fig 6B: Please explain how "relative expression" was assessed.

We included more detailed explanation in the methods section and in the legend of the figure (Page 22, line 587).

Fig 6G-I: Please explain the role of LPS in this assay and discuss the relevance of LPS during macrophage polarisation in the tumour microenvironment.

LPS stimulation has been widely used to polarize in vitro macrophages towards a pro-inflammatory M1 phenotype. We used LPS to show that sialic acids are not only involved in monocyte-to-macrophage differentiation but is also able to dampen macrophage activation and M1 polarization. The data of Figure 6G-I shows that sialic acid dendrimers alone induce upregulation of IL-10. However, if we look at its effect on inflammatory cytokines such as TNF α , we only observe production once stimulated with LPS.

Given that the polarization of TAMs towards a pro-inflammatory phenotype has been proposed as an anti-cancer therapy, our results show that this strategy can be affected by the presence of sialic acid in the tumour⁷.

We added this discussion in the manuscript to make clear why LPS was used (Page 14, lines 365-370).

Fig 7: Schematic only, no data shown. Thus, this figure should be combined with figure 6 or be move into supplement.

We added this figure into figure 6

Reviewer #4 (Remarks to the Author):

All raised aspects has been addressed. The paper 2 in current form is significantly improved and clear.

References

1. Haque A, Engel J, Teichmann SA and Lonnberg T, *A practical guide to single-cell RNA-sequencing for biomedical research and clinical applications*. Genome Med, 2017. **9**(1): p. 75.
2. Peng J, Sun BF, Chen CY, Zhou JY, Chen YS, Chen H, et al., *Single-cell RNA-seq highlights intratumoral heterogeneity and malignant progression in pancreatic ductal adenocarcinoma*. Cell Res, 2019.
3. Herring CA, Banerjee A, McKinley ET, Simmons AJ, Ping J, Roland JT, et al., *Unsupervised Trajectory Analysis of Single-Cell RNA-Seq and Imaging Data Reveals Alternative Tuft Cell Origins in the Gut*. Cell Syst, 2018. **6**(1): p. 37-51 e9.
4. Haghverdi L, Buttner M, Wolf FA, Buettner F and Theis FJ, *Diffusion pseudotime robustly reconstructs lineage branching*. Nat Methods, 2016. **13**(10): p. 845-8.
5. Azizi E, Carr AJ, Plitas G, Cornish AE, Konopacki C, Prabhakaran S, et al., *Single-Cell Map of Diverse Immune Phenotypes in the Breast Tumor Microenvironment*. Cell, 2018. **174**(5): p. 1293-1308 e36.
6. Sander J, Schmidt SV, Cirovic B, McGovern N, Papantonopoulou O, Hardt AL, et al., *Cellular Differentiation of Human Monocytes Is Regulated by Time-Dependent Interleukin-4 Signaling and the Transcriptional Regulator NCOR2*. Immunity, 2017. **47**(6): p. 1051-1066 e12.
7. Poh AR and Ernst M, *Targeting Macrophages in Cancer: From Bench to Bedside*. Front Oncol, 2018. **8**: p. 49.

REVIEWERS' COMMENTS

Reviewer #2 (Remarks to the Author):

The authors have done a reasonable job in addressing my comments/questions.

David T Ting MD
MGH Cancer Center

Reviewer #3 (Remarks to the Author):

Rodriguez et al., "Sialic acids in pancreatic cancer cells drive tumour-1 associated macrophage differentiation via Siglec-7 and Siglec-9"

This is a revised version of a previously submitted manuscript. In the rebuttal letter, the authors have addressed well most of my initial concerns. Overall, the quality of the experimental data has improved, while the methods how computational analyses were performed are better described.

Following comments remain:

Line 138: articulate that the findings relate to listed pancreatic cancer cell lines.

Line 146: specify that shorter survival relates to pancreatic cancer patients.

Line 291: Please amend statement "...to facilitate immune evasion in PDAC". In the current version, the manuscript doesn't address the immunosuppressive functions of macrophages. In fact, it would strengthen this manuscript to add some functional in vitro data to show that sialic acid stimulation increases the immunosuppressive capacity of macrophages in terms of T cell or NK cell activation.

Figure 5F/G: Depletion of Sia-activating enzyme CMAS results in reduced IL-10 secretion in co-culture assays. Does CMAS depletion affect M-CSF or IL-10 expression in pancreatic cancer cells which might affect macrophage differentiation or IL-10 levels in the co-culture system?

Fig 6C: Is the significant increase of IL-10 production in response to sialic acid stimulation depended on the absence of M-CSF? Please check figure and labelling since current version shows no significant effect of sialic acid on IL-10 secretion in M-CSF treated group.

Reviewer #4 (Remarks to the Author):

Adjustments significantly improved the manuscript. Particularly, data are now realistically presented.

Reviewer #2 (Remarks to the Author):

The authors have done a reasonable job in addressing my comments/questions.

David T Ting MD

MGH Cancer Center

Response from authors:

We would like to thank Dr Ting for his comments and valuable contributions during the review of this manuscript.

Reviewer #3 (Remarks to the Author):

Rodriguez et al., "Sialic acids in pancreatic cancer cells drive tumour-1 associated macrophage differentiation via Siglec-7 and Siglec-9"

This is a revised version of a previously submitted manuscript. In the rebuttal letter, the authors have addressed well most of my initial concerns. Overall, the quality of the experimental data has improved, while the methods how computational analyses were performed are better described.

Following comments remain:

Line 138: articulate that the findings relate to listed pancreatic cancer cell lines.

We have added that information in the main text (Page 6, line 139).

Line 146: specify that shorter survival relates to pancreatic cancer patients.

We have added that information in the main text (Page 6, line 146).

Line 291: Please amend statement "...to facilitate immune evasion in PDAC". In the current version, the manuscript doesn't address the immunosuppressive functions of macrophages. In fact, it would strengthen this manuscript to add some functional in vitro data to show that sialic acid stimulation increases the immunosuppressive capacity of macrophages in terms of T cell or NK cell activation.

We agree with the reviewer and deleted that phrase of the main text (Page 11, line 292). We consider the assays suggested valuable, but these are out of the scope of the present manuscript, and would take considerable time (months) to perform these.

Figure 5F/G: Depletion of Sia-activating enzyme CMAS results in reduced IL-10 secretion in co-culture assays. Does CMAS depletion affect M-CSF or IL-10 expression in pancreatic cancer cells which might affect macrophage differentiation or IL-10 levels in the co-culture system?

No significant differences were observed in the expression of IL-10, M-CSF, IL-6 and TNF α between the CMAS KO and WT pancreatic cancer cell lines.

Fig 6C: Is the significant increase of IL-10 production in response to sialic acid stimulation depended on the absence of M-CSF? Please check figure and labelling since current version shows no significant effect of sialic acid on IL-10 secretion in M-CSF treated group.

Indeed, the Figure 6C shows that sialic acid dendrimer increases IL-10 secretion by monocytes only when M-CSF is absent. However, sialic acid dendrimers can also increase IL-10 production by M-CSF differentiated macrophages, as shown in Figure 6 I. We have changed the manuscript (Page 10, line 274).

Reviewer #4 (Remarks to the Author):

Adjustments significantly improved the manuscript. Particularly, data are now realistically presented.

Response from authors:

We would like to thank the reviewer for his comments and valuable contributions during the review of this manuscript.